# A rapid phylogeny-based method for accurate community profiling of large-scale metabarcoding datasets

Lenore Pipes[1]*, Rasmus Nielsen[1,2]

[1]Department of Integrative Biology, University of California, Berkeley, Berkeley, United States; [2]GLOBE Institute, University of Copenhagen, Copenhagen, Denmark

**Abstract** Environmental DNA (eDNA) is becoming an increasingly important tool in diverse scientific fields from ecological biomonitoring to wastewater surveillance of viruses. The fundamental challenge in eDNA analyses has been the bioinformatical assignment of reads to taxonomic groups. It has long been known that full probabilistic methods for phylogenetic assignment are preferable, but unfortunately, such methods are computationally intensive and are typically inapplicable to modern next-generation sequencing data. We present a fast approximate likelihood method for phylogenetic assignment of DNA sequences. Applying the new method to several mock communities and simulated datasets, we show that it identifies more reads at both high and low taxonomic levels more accurately than other leading methods. The advantage of the method is particularly apparent in the presence of polymorphisms and/or sequencing errors and when the true species is not represented in the reference database.

## Editor's evaluation

This important work presents a novel tool for performing phylogenetic assignment of DNA sequences. The manuscript is convincing, and the authors perform a standard benchmark experiment against current state-of-the-art tools using real + simulated datasets to demonstrate where the novel tool stands in the context of existing methods. This paper will be of great interest to bioinformaticians and evolutionary biologists interested in massively-scalable phylogenetic assignment.

*For correspondence:
lpipes@berkeley.edu

Competing interest: The authors declare that no competing interests exist.

## Introduction

In the past 10 years, metabarcoding and metagenomics based on DNA sequencing and subsequent taxonomic assignment have become an important approach for understanding diversity and community organization at many taxonomic levels. This has led to the publication of more than 80 taxonomic classification methods (*Gardner et al., 2019*). There are three major strategies in classification methods: (1) composition-based, which do not align sequences but extract compositional features (e.g., kmers) to build models of probabilistic taxonomic inclusion; (2) alignment-based, which rely on alignments to directly compare query sequences to reference sequences but do not use trees; and (3) phylogenetic-based, which rely on a phylogenetic tree reconstruction method, in addition to alignments, to perform a placement of the query onto the tree. As a trade-off between speed and precision for processing next-generation sequencing (NGS) data, the vast majority of recent classification methods have either relied on alignment-based or composition-based strategies.

Composition-based tools reduce the reference database by indexing compositional features such as kmers for a rapid search of the database. These methods require an exact match between the kmer in the query sequence and the kmer in the reference database. As a result of hash indexing

**Figure 1.** Species assignment in alignment-based methods (**A**) vs. Tronko (**B**). In Tronko, scores are calculated for all nodes in the tree based on the query's global alignment to the best BWA-MEM hit. The query is assigned to the lowest common ancestor (LCA) of the highest scoring nodes within the cut-off threshold. See *Figure 1—figure supplement 1* for more details regarding using multiple trees.

The online version of this article includes the following figure supplement(s) for figure 1:

**Figure supplement 1.** Workflow of iterative partitioning procedure.

**Figure supplement 2.** Comparisons of Tronko with pplacer and APPLES-2 using a database of 200, 400, 600, 800, 1000, 1200, 1400, and 1600 reference sequences.

of kmers, kraken2 (*Wood et al., 2019*), for example, can classify >1 million reads within 1 min using the entire Geengenes or SILVA databases (*Lu and Salzberg, 2020*). Alignment-based tools use a fast local aligner such as BLAST (*Altschul et al., 1990*) to pairwise align queries to the reference database and define a score based on sequence similarity in the alignment between the read and reference sequence. However, alignment-based methods can be many orders of magnitude slower than composition-based tools since datasets with >10 million reads require weeks of BLASTN running time (*Ainsworth et al., 2017*). In both composition-based tools and alignment-based tools, a lowest common ancestor (LCA) algorithm is then typically used to assign at different taxonomic levels (*Figure 1A*). LCA works by assigning to the smallest possible clade that include all matches with a similarity less than the specified cut-off.

Phylogenetic placement methods place a query sequence onto a phylogenetic tree of reference sequences. This placement requires a full multiple sequence alignment (MSA) of the reference sequences and a subsequent estimation of a phylogenetic tree. However, large datasets with high rates of evolution are hard to align accurately (*Sievers et al., 2011*) and phylogeny estimation methods produce poor trees when MSAs are not of high quality (*Kapli et al., 2020*). Furthermore, phylogenetic placement tends to be computationally demanding as both running time and memory usage scale linearly with the size of the reference database (*Balaban et al., 2020*). Even for reference databases that contain sequences as few as 1600 sequences, assignment for a single query using the most cited

phylogenetic placement method, *pplacer* (*Matsen et al., 2010*), takes more than 7 min and requires over 10 GB of RAM (on a Dell PowerEdge server with 32 CPU threads and 512 GB of RAM). At this rate, a reference database that contains a metabarcode such as cytochrome oxidase 1 (COI) that has at least 1.5 million reference sequences, assigning just a single query would require 20.9 hr and 2.37 TB RAM. Scaling the query size to millions of queries would therefore be computationally intractable.

To address these challenges, the most recent implementations of phylogeny-based methods (*Barbera et al., 2019*) rely on reference database reduction techniques (i.e., using only representative taxa or consensus sequences for a sparse backbone tree) to handle the large amount of data that is routinely produced. Often a single species is selected to represent an entire clade (*Czech et al., 2022*). While this reduces the computational cost, it also reduces the granularity, and potentially the accuracy, of the assignments. As a trade-off between speed and precision, the vast majority of recent classification methods are either alignment-based or composition-based approaches (*Hleap et al., 2020*) since phylogeny-based methods have not scaled to handle the entirety of the rapidly growing reference databases of genome markers and the increasingly large amounts of NGS data.

Here, we describe a new method for phylogenetic placement, implemented in the program 'Tronko' (https://github.com/lpipes/tronko, *Mapper and Pipes, 2024*), the first phylogeny-based taxonomic classification method designed to truly enable the use of modern-day reference databases and NGS data. The method is based on approximating the phylogenetic likelihood calculation by (1) only allowing the edge connecting the reference sequence to the tree to join at existing nodes in the tree and then (2) approximating the likelihood using a probabilistically weighted mismatch score based on pre-calculated fractional likelihoods stored in each node (see 'Methods'). We argue that (2) approximates the full maximized likelihood assignment without requiring any numerical maximization under the approximating assumption that the read joins the tree in an existing node with a zero-length branch. The approximation is equivalent to calculating the expected average mismatch to each node in the phylogeny. The assignment method in Tronko uses the LCA criteria but, unlike composition-based and alignment-based approaches (*Figure 1A*), takes advantage of fractional likelihoods stored in all nodes of the tree with a cut-off that can be adjusted from conservative to aggressive (*Figure 1B*). In the simplest case, when the reference sequences form a single tree, Tronko uses a pre-calculated MSA, the phylogenetic tree based on the MSA, and pre-calculated posterior probabilities, which are proportional to the fractional likelihoods. However, in more typical cases, when a single tree/MSA is unsuitable for analyses, as the reference sequences encompass increasingly divergent species as well as an increasing volume of sequences, we present a fully customizable divide-and-conquer method for reference database construction that is based on dividing reference sequences into phylogenetic subsets that are realigned and with local trees re-estimated.

The construction of the database, MSAs, and trees facilitates fast phylogenetic assignment. The assignment algorithm then proceeds by (1) a BWA-MEM (*Li and Durbin, 2009*) search on all sequences in the database, (2) a pairwise sequence alignment between the query and the top hit in each alignment subset containing a BWA-MEM hit using either the Needleman–Wunsch algorithm (*Needleman and Wunsch, 1970*) or the wavefront alignment algorithm (*Marco-Sola et al., 2021*), and (3) a calculation of a score based on the approximate likelihood for each node in subsets with a BWA-MEM hit. An additional LCA assignment for all subsets can then be applied to summarize the results. For full details, see 'Methods'.

## Results

To compare the new method (Tronko) to previous methods, we constructed reference databases for COI and 16S for common amplicon primer sets using CRUX (*Curd et al., 2019*; sSee 'Methods' for the exact primers used). We first compared Tronko to pplacer and APPLES-2 for reference databases containing a reduced amount of sequences (<1600 sequences) to compare the speed and memory requirements with comparable phylogenetic-based assignment methods. Tronko shows speed-ups >20 times over pplacer, with a vastly reduced memory requirement illustrating the computational advantage of the approximations in Tronko (*Figure 1—figure supplement 2*). Tronko demonstrates a speed-up >2 times over APPLES-2 with a similar memory footprint. In terms of accuracy, all methods had a 100% true-positive rate at the species level. Additionally, in terms of the species assignment rate (the percentage of queries that were assigned at the species level), Tronko assigns the most queries.

Next, in addition to pplacer and APPLES-2, we evaluated Tronko's performance to kmer-based kraken2 (*Wood et al., 2019*), which previously has been argued to have the lowest false-positive rate (*Lu and Salzberg, 2020*), and two other popular alignment-based methods: MEGAN (*Huson et al., 2007*) and metaphlan2 (*Truong et al., 2015*). We used two types of cross-validation tests: leave-one-species-out and leave-one-individual-out analyses. The leave-one-species-out test involves removing an entire species from the reference database, simulating NGS reads from that species, and then attempting to assign those reads with that species missing from the database. The leave-one-individual-out test involves removing a single individual from the reference database, simulating NGS reads from that individual, and then attempting to assign those reads with that individual missing from the database. In both tests, singletons (i.e., cases in which only one species was present in a genera or cases in which only one individual represented a species) were exempt from the tests.

We performed a leave-one-species-out test comparing Tronko (with LCA cut-offs for the score of 0, 5, 10, 15, and 20 with both Needleman–Wunsch alignment and wavefront alignment) to kraken2, metaphlan2, and MEGAN for 1467 COI sequences from 253 species from the order Charadriiformes using 37,515 (150 bp × 2) paired-end sequences and 768,807 single-end sequences (150 bp and 300 bp in length) using 0, 1, and 2% error/polymorphism (*Figure 2*). We use the term 'error/poly-morphism' to represent a simulated change in nucleotide that can be either an error in sequencing or a polymorphism. We display confusion matrices to display the clades in which each method has an incorrect assignment (*Figure 3*). See *Figure 2—figure supplement 1* for results with the wavefront alignment algorithm (*Marco-Sola et al., 2021*).

Using leave-one-species-out and simulating reads (both paired-end and single-end) with a 0–2% error (or polymorphism), Tronko detected the correct genus more accurately than the other methods even when using an aggressive cut-off (i.e., when cut-off 0) (*Figure 3F*). Using 150 bp paired-end reads with 1% error, Tronko had a misclassification rate of only 9.8% with a recall rate of 70.1% at the genus level using a cut-off set to 15 while kraken2, MEGAN, and metaphlan2 had misclassification rates of 33.5, 10.0, and 27.7%, respectively, with recall rates of 90.6, 52.1, and 95.0% (see *Figure 2B*). Tronko had a lower misclassification rate relative to the recall rate out of all methods for 150 bp × 2 paired-end reads with 0% error/polymorphism (*Figure 2A*), 1% error/polymorphism (*Figure 2B*), and 2% error/polymorphism (*Figure 2* and *Figure 3D–I*), for 150 bp reads with 0% error/polymorphism (*Figure 2D*), 1% error/polymorphism (*Figure 2E*), and 2% error/polymorphism (*Figure 2F*), and for 300 bp reads with 0% error/polymorphism (*Figure 2G*), 1% error/polymorphism (*Figure 2H*), and 2% error/polymorphism (*Figure 2I*). See 'Methods' for definitions of recall and misclassification rates. Tronko also accurately assigned genera from the Scolopacidae family (top left of matrices in *Figure 3*) using Needleman–Wunsch with a cut-off of 10 compared to kraken2, meta-phlan2, and pplacer.

Next, we performed a leave-one-individual-out test for the same COI reference sequences using 746,352 single-end reads and 36,390 paired-end reads (*Figure 4*, *Figure 4—figure supplement 1F–J*). See *Figure 4—figure supplement 2* for results with wavefront alignment algorithm (*Marco-Sola et al., 2021*). Using single-end reads of lengths 150 bp and 300 bp, Tronko has a lower misclassification rate and higher recall rate than kraken2, metaphlan2, and MEGAN. Using 150 bp paired-end reads with 0% error (*Figure 4D*), Tronko had a misclassification rate at only 0.1% with a recall rate of 58.6% at the species level using a cut-off set to 10 while kraken2, MEGAN, and metaphlan2 had misclassification rates of 1.5, 0.1, and 11.0%, respectively, with recall rates of 85.4, 60.7, and 98.14%. Both metaphlan and kraken2 have a number of mis-assignments within the family of Laridae (see blue points across the diagonal in *Figure 4—figure supplement 1A and B*) and Tronko is able to accurately assign species within this family or assign at the genus or family level. We also observe that for increasing error rates, kraken2 and metaphlan2 have a substantial increase in misclassification rate. We believe it would be slightly misleading to display results for pplacer and APPLES-2 here due to the lack of an implementation to calculate the LCA on similar likelihoods. See *Figure 4—figure supplement 2* for results for pplacer and APPLES-2 along with wavefront alignment algorithm.

In order to replicate real-world scenarios, we added a leave-one-species-out test using 16S from 2323 bacterial species and 5000 individual sequences (*Figure 5*). We selected the sequences for the 16S dataset by grouping the sequences by the class level in a random order, rotating that order, and randomly selecting an individual sequence from each group. We then simulated sequencing reads from the dataset simulating 21,947,613 single-end reads (150 bp and 300 bp in length) as well as

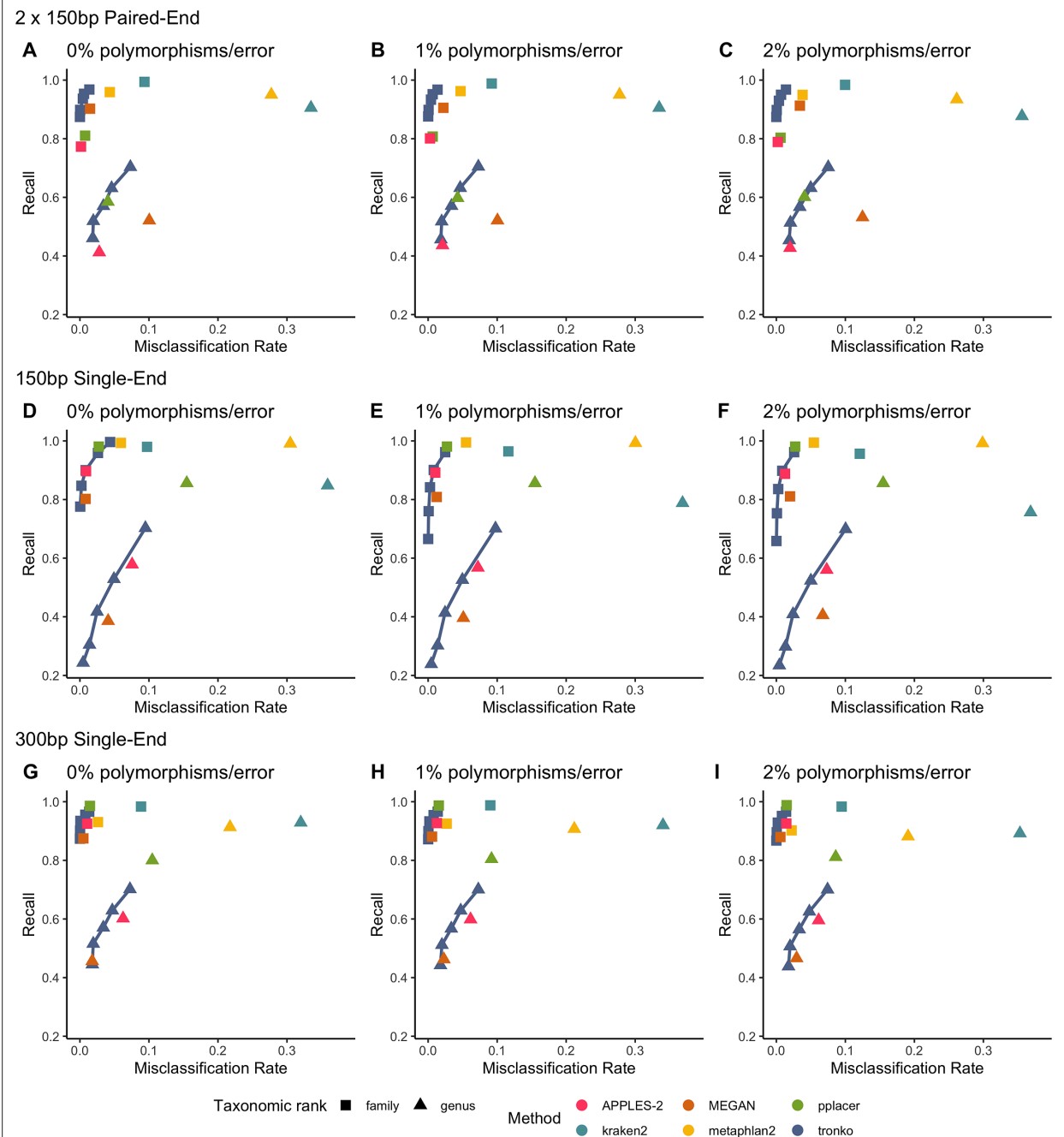

**Figure 2.** Recall vs. misclassification rates using leave-one-species-out analysis of the order Charadriiformes (cytochrome oxidase 1 [COI] metabarcode) with paired-end 150 bp × 2 reads with 0% (**A**), 1% (**B**), and 2% (**C**) error/polymorphism, single-end 150 bp reads with 0% (**D**), 1% (**E**), and 2% (**F**) error/ polymorphism, and single-end 300 bp reads with 0% (**G**), 1% (**H**), and 2% (**I**) error/polymorphism using kraken2, metaphlan2, MEGAN, pplacer, and APPLES-2, and Tronko with cut-offs of 0, 5, 10, 15, and 20 using the Needleman–Wunsch alignment (solid line). See *Figure 2—figure supplement 2* for results using different combinations of aligners and tree estimation methods.

The online version of this article includes the following figure supplement(s) for figure 2:

**Figure supplement 1.** Recall vs. misclassification rates using leave-one-species-out analysis for the order Charadriiformes (cytochrome oxidase 1 [COI] metabarcode) with paired-end 150 bp × 2 reads with 0% (**A**), 1% (**B**), and 2% (**C**) error/polymorphism, single-end 150 bp reads with 0% (**D**), 1% (**E**), and 2% (**F**) error/polymorphism, and single-end 300 bp reads with 0% (**G**), 1% (**H**), and 2% (**I**) error/polymorphism using kraken2, metaphlan2, MEGAN, pplacer, APPLES-2, and Tronko with cut-offs of 0, 5, 10, 15, and 20 using the Needleman–Wunsch alignment (solid line) and wavefront alignment (dashed line).

*Figure 2 continued on next page*

*Figure 2 continued*

**Figure supplement 2.** Recall vs. misclassification rates using leave-one-species-out analysis for the order Charadriiformes (cytochrome oxidase 1 [COI] metabarcode) with paired-end 150 bp × 2 reads with 2% error/polymorphism using Tronko with cut-offs of 0, 5, 10, 15, and 20 and different combinations of tree estimation methods and aligners.

21,478,738 paired-end 150 bp × 2 reads. In all simulations at both the genus and family levels, Tronko had a higher recall and a lower misclassification rate than all other methods. The simulations for 300 bp single-end reads are not directly comparable to the 150 bp single-end or paired-end reads because only 105 missing-out tests out of 2310 were able to be performed because most reference sequences were <300 bp in length. We only display the results for 300 bp single-end reads for APPLES-2 in the supplement as we believe the results are not a good representation of the method. See *Figure 5—figure supplement 1* for results for APPLES-2 using 300 bp single-end reads, along with results using the wavefront alignment algorithm. Additionally, we tested the use of hmmer or MAFFT for alignments with APPLES-2 and pplacer (*Figure 5—figure supplement 2*), and we did not observe any substantial difference with the choice of alignment.

We then compared Tronko's performance to kraken2, MEGAN, and metaphlan2 using mock communities for both 16S (*Schirmer et al., 2015*; *Gohl et al., 2016*) and COI markers (*Braukmann et al., 2019*; *Figure 6*). We did not compare mock community data to pplacer and APPLES-2 because we were unsuccessful in building a full MSA for our 16S and COI reference databases. Tronko also relies on sequence alignments, but as described in 'Methods', they can be handled by dividing sequences into clusters in the Tronko pipeline. For 16S, we used three different mock community datasets. We used 1,054,868 2 × 300 bp Illumina MiSeq sequencing data from a mock community consisting of 49 bacteria and 10 archaea species from *Schirmer et al., 2015*, 54,930 2× 300 bp Illumina MiSeq sequencing data from a mock community consisting of 14 bacteria species from *Lluch et al., 2015*, and 206,696 2 × 300 bp Illumina MiSeq sequencing data from a mock community of 20

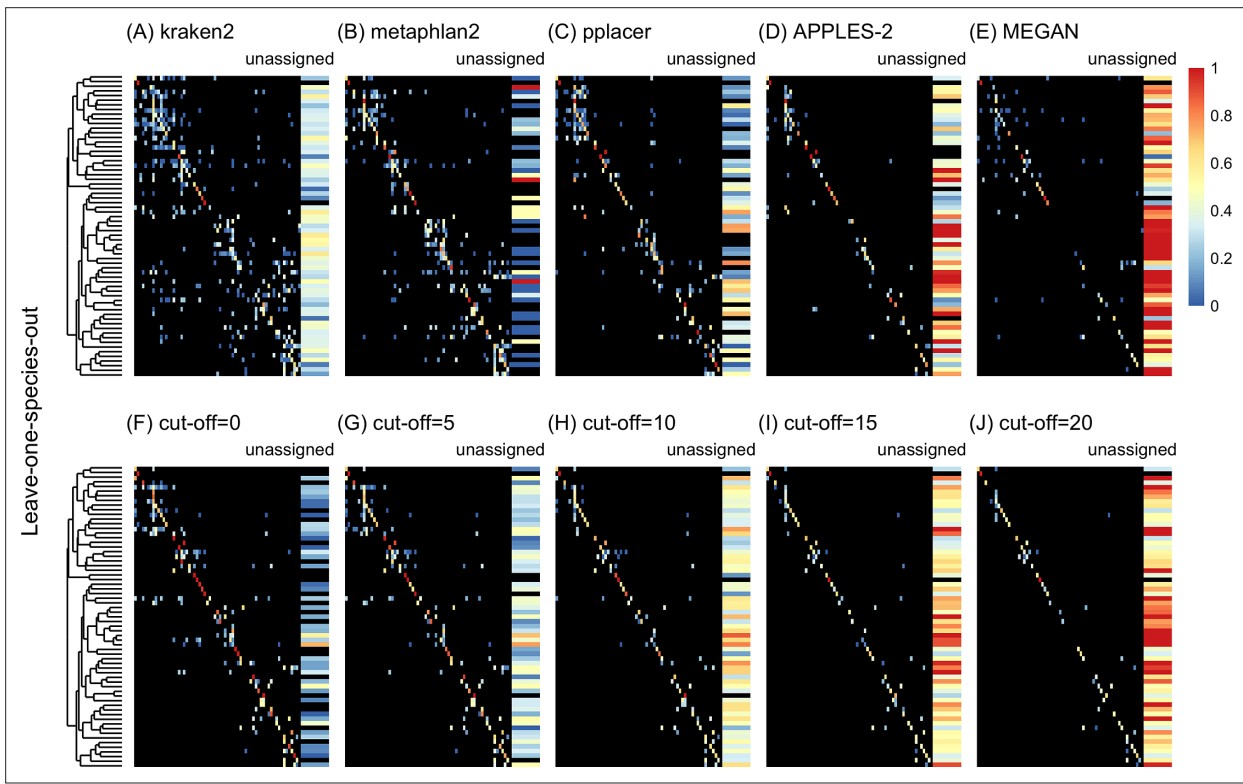

**Figure 3.** Confusion matrices at the genus level of the order Charadriiformes (cytochrome oxidase 1 [COI] metabarcode) using the leave-one-species-out analysis with paired-end 150 bp × 2 reads with 2% error/polymorphism using kraken2 (**A**), metaphlan2 (**B**), pplacer (**C**), APPLES-2 (**D**), MEGAN (**E**), and Tronko using the Needleman–Wunsch alignment (NW) for cut-offs 0 (**F**), 5 (**G**), 10 (**H**), 15 (**I**), and (**J**) 20. Unassigned column contains both unassigned queries and queries assigned to a lower taxonomic level. Phylogenetic tree represents ancestral sequences at the genus level.

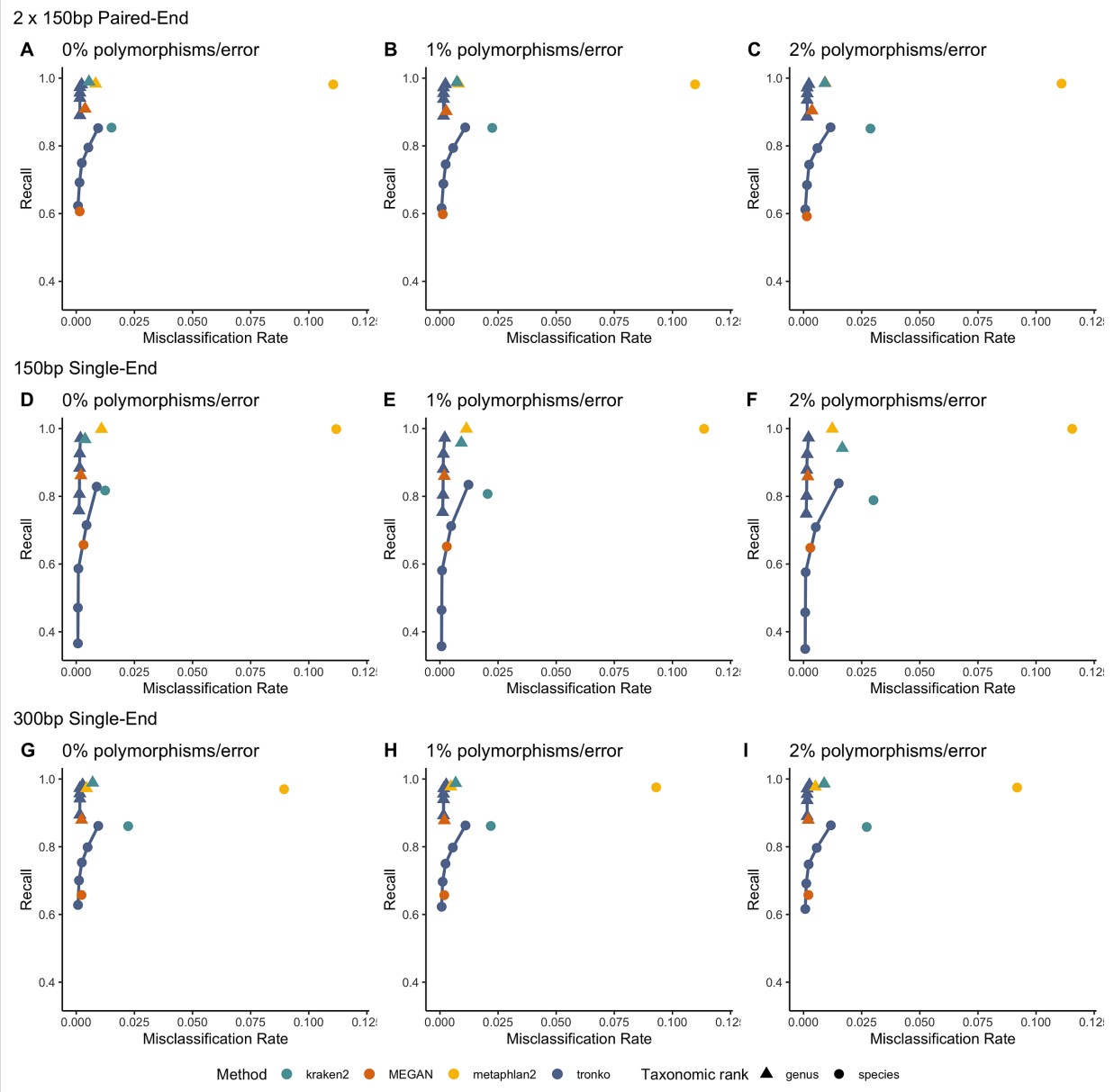

**Figure 4.** Recall vs. misclassification rates using leave-one-individual-out analysis for the order Charadriiformes (cytochrome oxidase 1 [COI] metabarcode) with paired-end 150 bp × 2 reads with 0% (**A**), 1% (**B**), and 2% (**C**) error/polymorphism, single-end 150 bp reads with 0% (**D**), 1% (**E**), and 2% (**F**) error/polymorphism, and single-end 300 bp reads with 0% (**G**), 1% (**H**), and 2% (**I**) error/polymorphism using kraken2, metaphlan2, MEGAN, pplacer, APPLES-2, and Tronko with cut-offs of 0, 5, 10, 15, and 20 using the Needleman–Wunsch alignment (solid line).

The online version of this article includes the following figure supplement(s) for figure 4:

**Figure supplement 1.** Confusion matrices at the species level of the order Charadriiformes using the leave-one-individual-out analysis with paired-end 150 bp × 2 reads with 2% error/polymorphism using kraken2 (**A**), metaphlan2 (**B**), pplacer (**C**), APPLES-2 (**D**), MEGAN (**E**), and Tronko using the Needleman–Wunsch alignment (NW) for cut-offs 0 (**F**), 5 (**G**), 10 (**H**), 15 (**I**), and (**J**) 20.

**Figure supplement 2.** Recall vs. misclassification rates using leave-one-individual-out analysis for the order Charadriiformes (cytochrome oxidase 1 [COI] metabarcode) with paired-end 150 bp × 2 reads with 0% (**A**), 1% (**B**), and 2% (**C**) error/polymorphism, single-end 150 bp reads with 0% (**D**), 1% (**E**), and 2% (**F**) error/polymorphism, and single-end 300 bp reads with 0% (**G**), 1% (**H**), and 2% (**I**) error/polymorphism using kraken2, metaphlan2, MEGAN, pplacer, APPLES-2, and Tronko with cut-offs of 0, 5, 10, 15, and 20 using the Needleman–Wunsch alignment (solid line) and wavefront alignment (dashed line).

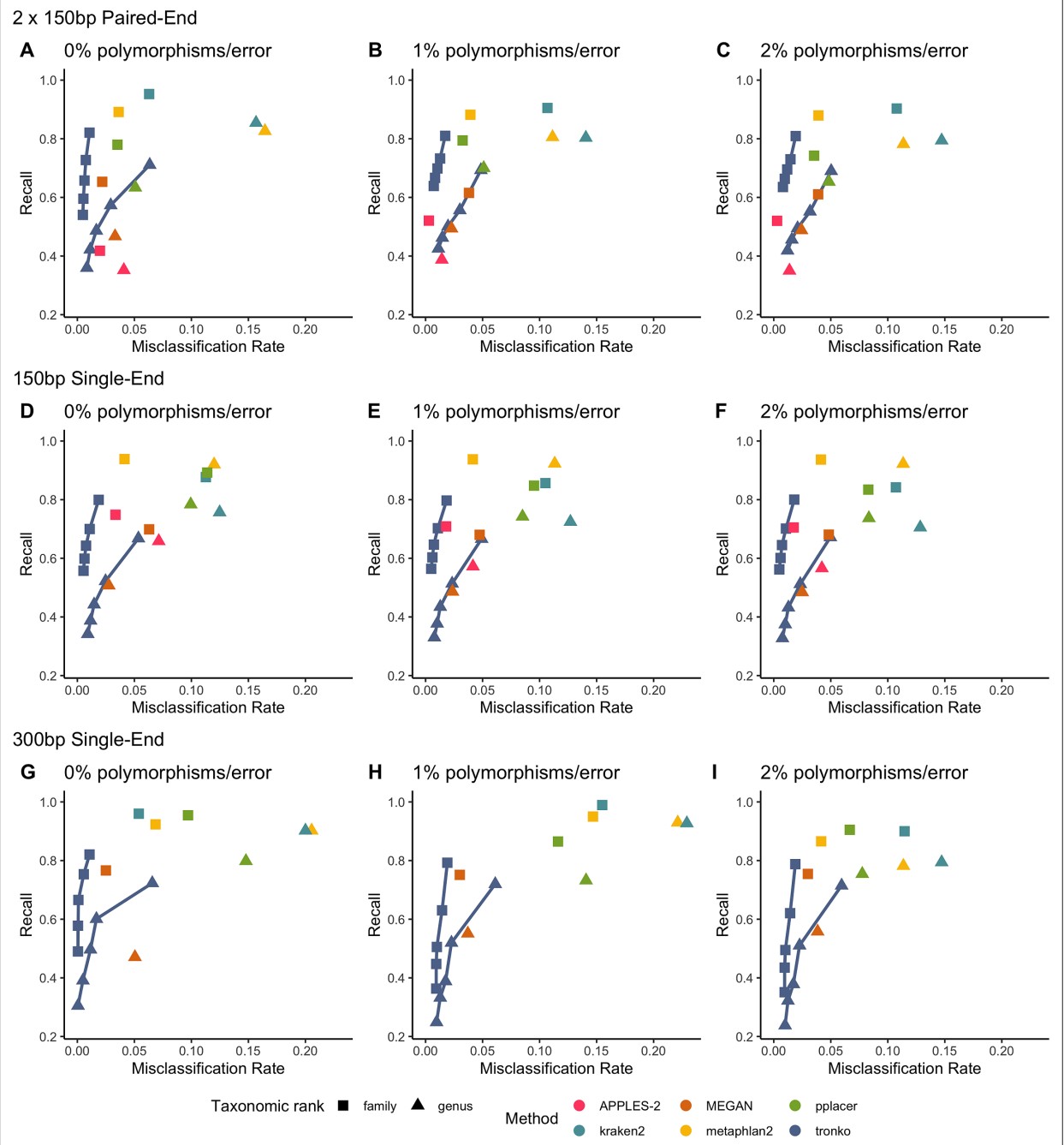

**Figure 5.** Recall vs. misclassification rates using leave-one-species-out analysis with bacteria species (16S metabarcode) with paired-end 150 bp × 2 reads with 0% (**A**), 1% (**B**), and 2% (**C**) error/polymorphism, single-end 150 bp reads with 0% (**D**), 1% (**E**), and 2% (**F**) error/polymorphism, and single-end 300 bp reads with 0% (**G**), 1% (**H**), and 2% (**I**) error/polymorphism using kraken2, metaphlan2, MEGAN, pplacer, APPLES-2 and Tronko with cut-offs of 0, 5, 10, 15, and 20 using the Needleman–Wunsch alignment (solid line).

The online version of this article includes the following figure supplement(s) for figure 5:

**Figure supplement 1.** Recall vs. misclassification rates using leave-one-species-out analysis with bacteria species (16S metabarcode) with paired-end 150 bp × 2 reads with 0% (**A**), 1% (**B**), and 2% (**C**) error/polymorphism, single-end 150bp reads with 0% (**D**), 1% (**E**), and 2% (**F**) error/polymorphism, and single-end 300bp reads with 0% (**G**), 1% (**H**), and 2% (**I**) error/polymorphism using kraken2, metaphlan2, MEGAN, pplacer, APPLES-2, and Tronko with cut-offs of 0, 5, 10, 15, and 20 using the Needleman–Wunsch alignment (solid line) and wavefront alignment (dashed line).

**Figure supplement 2.** Recall vs. misclassification rates using leave-one-individual-out analysis for bacterial species (16S metabarcode) with paired-end 150 bp × 2 reads with 0% (**A**), 1% (**B**), and 2% (**C**) error/polymorphism using kraken2, metaphlan2, MEGAN, pplacer + hmmer, pplacer + mafft, APPLES-2 + hmmer, APPLES-2 + mafft, and Tronko with cut-offs of 0, 5, 10, 15, and 20 using the Needleman–Wunsch alignment (solid line).

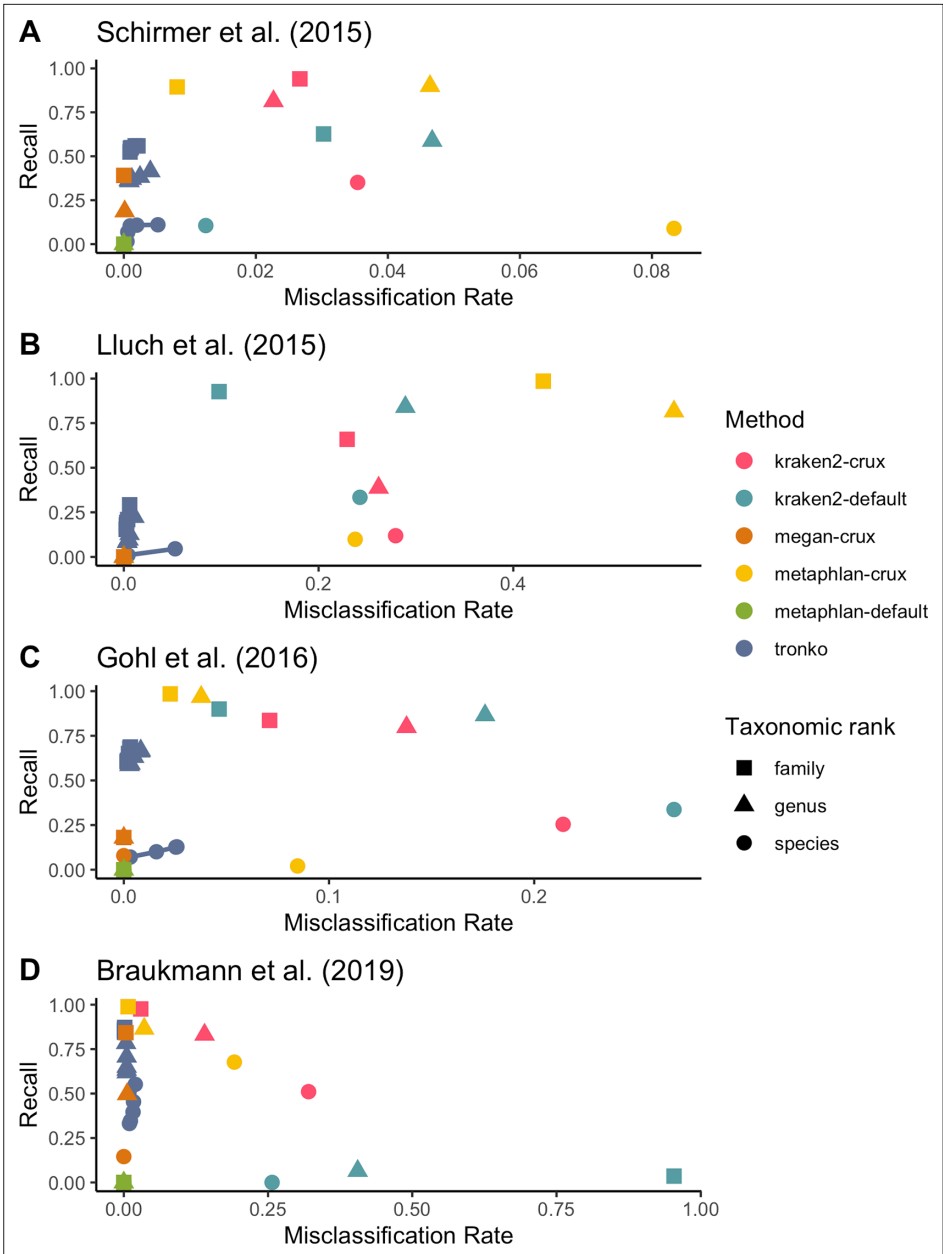

**Figure 6.** Recall vs. misclassification rates using mock communities from *Schirmer et al., 2015* (**A**), *Lluch et al., 2015* (**B**), *Gohl et al., 2016* (**C**), and *Braukmann et al., 2019* (**D**) using both Needleman–Wunsch and wavefront alignment algorithms. Figures with smaller misclassification rates on the x-axis are available for *Schirmer et al., 2015*, *Lluch et al., 2015*, *Gohl et al., 2016*, *Braukmann et al., 2019* in *Figure 6—figure supplements 1, 2*, and *4*, respectively.

The online version of this article includes the following figure supplement(s) for figure 6:

**Figure supplement 1.** Close-up of *Figure 6A*.

**Figure supplement 2.** Close-up of *Figure 6A*.

**Figure supplement 3.** Close-up of *Figure 6B*.

**Figure supplement 4.** Close-up of *Figure 6C*.

evenly distributed bacterial species from *Gohl et al., 2016*. For the data from *Schirmer et al., 2015*, at the species level, Tronko had a <0.6% misclassification rate at every cut-off with a recall rate of 11.0% at cut-off 0 (*Figure 6A*; see *Figure 6—figure supplement 1* for plot without outliers). kraken2 had a misclassification rate of 1.2% with a recall rate of 10.6% when using its default database, and

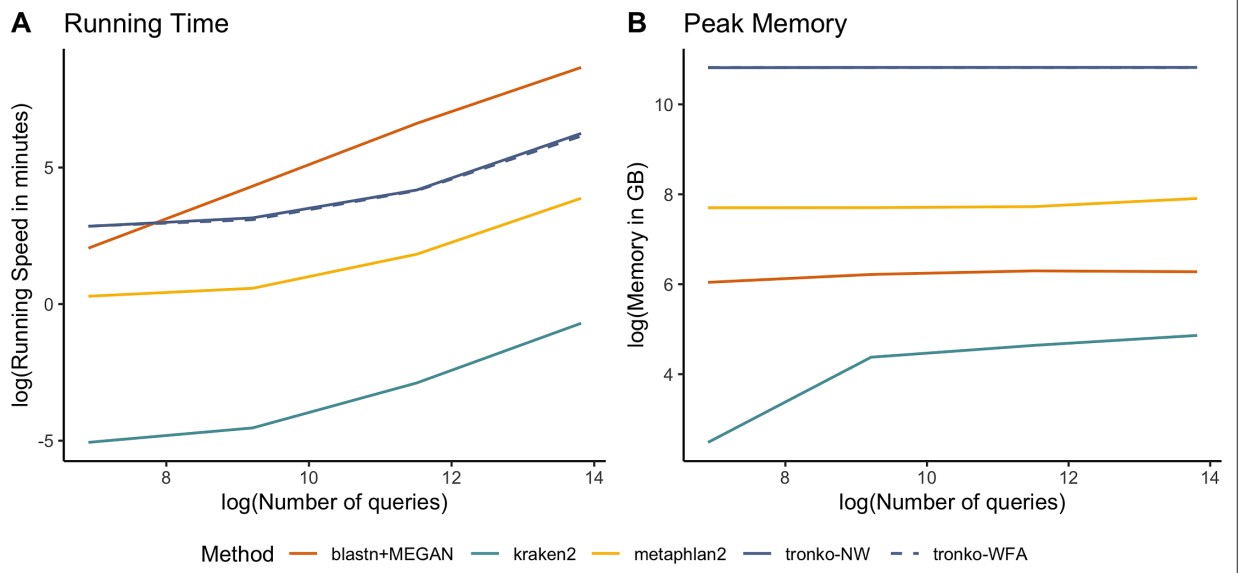

**Figure 7.** Comparisons of running time (**A**) and peak memory (**B**) using 100, 1000, 10,000, 100,000, and 1,000,000 queries for Tronko, blastn + MEGAN, kraken2, and metaphlan2 using the cytochrome oxidase 1 (COI) reference database. NW: Needleman–Wunsch; WFA: wavefront alignment.

a misclassification rate of 3.5% and a recall rate of 35.1% when using the same reference sequences as Tronko. metaphlan2 did not have any assignments at the species, genus, or family level using the default database, and it had an 8.3% misclassification and 8.9% recall rate at the species level when using the same reference sequences as Tronko. MEGAN had a recall rate of 0.2% and a misclassification rate of 0% at the species level.

For the data from *Lluch et al., 2015*, at the genus level, Tronko had a misclassification rate of 0.6% and a recall rate of 22.3% using a cut-off of 0, while all other methods had a misclassification rate of >8% (see *Figure 6—figure supplement 2* for a close-up of the rates).

For the data from *Gohl et al., 2016*, at the species level, Tronko had a <2.6% misclassification rate at every cut-off with a recall rate of 12.8% at cut-off 0 (*Figure 6C*; see *Figure 6—figure supplement 3* for plot without outliers). kraken2 had a misclassification rate of 26.8% and recall rate of 33.7% when using its default database, and a misclassification rate of 21.4% and recall rate of 25.4% when using the same reference sequences as Tronko. metaphlan2 did not have any assignments at the species, genus, or family level using the default database, and it had an 8.5% misclassification and 2.1% recall rate at the species level when using the same reference sequences as Tronko. MEGAN had a misclassification rate of 0% and a recall rate of 4.4% at the species level.

For COI, we used a dataset from *Braukmann et al., 2019* which consists of 646,997 2 × 300 bp Illumina MiSeq sequencing data from 374 species of terrestrial arthropods, which is the most expansive mock community dataset that we used. At the genus level, Tronko had a misclassification rate of <0.5% with a recall rate of 78.3% at the cut-off of 0 (*Figure 6D*; see *Figure 6—figure supplement 4* for plot without outliers). With the default database, kraken2 had a misclassification rate of 40.5% with a recall rate of 6.5%. With the same reference sequences as Tronko, kraken2 still had a misclassification of 14.0% with a recall rate of 83.1%. metaphlan2 had a misclassification rate of 3.5% with a recall of 86.4% with the same reference sequences as Tronko while the default database failed to assign any reads. MEGAN had a 15.0% recall and 0% misclassification rate at the species level and a 49.9% recall and 0.5% misclassfication rate at the genus level.

We compared Tronko with kraken2, metaphlan2, and MEGAN (using BLAST as the aligner) for running time (*Figure 7A*) and peak memory (*Figure 7B*) using 100, 1000, 10,000, 100,000, and 1,000,000 sequences using the COI reference database. Unsurprisingly, kraken2 had the fastest running time followed by metaphlan2, but MEGAN had a substantially slower running time than all methods. Tronko was able to assign 1,000,000 queries in ~8 hr with the choice of aligner being negligible. Tronko had the highest peak memory (~50 GBs) as it stores all reference sequences, their

trees, and their posterior probabilities in memory. We note that for very large databases, the memory requirements can, in theory, be reduced by processing different alignment subsets sequentially.

## Discussion

Both leave-one-species-out and leave-one-individual-out simulations show that Tronko recovers the correct taxonomy with higher probability than competing methods and represents a substantial improvement over current assignment methods. The advantage of Tronko comes from the use of limited full-sequence alignments and the use of phylogenetic assignment based on a fast approximation to the likelihood.

We evaluate Tronko using different cut-offs representing different trade-offs between recall and misclassification rate, thereby providing some guidance to users for choice of cut-off. We note that in most cases the other methods evaluated here fall within the convex hull of Tronko, showing that Tronko dominates those methods, and in no cases do other methods fall above the convex hull of Tronko. However, in some cases, other methods are so conservative, or anti-conservative, that a direct comparison is difficult. For example, when using single-end 300 bp reads (*Figure 4G–I*), MEGAN has assignment rates that are so low that a direct comparison is difficult.

Among the methods compared here, kraken2 is clearly the fastest (*Figure 7A*). However, it generally also has the worst performance with a higher misclassification rate than other methods, especially in the leave-one-species-out simulations (*Figure 2*).

Both metaphlan2 and MEGAN tend to fall within the convex hull of Tronko. Typically, metaphlan2 assigns much more aggressively, and therefore, has both a recall and misclassification rate that is much higher than MEGAN, which assigns very conservatively. We also note that the computational speed of MEGAN is so low that it, in some applications, may be prohibitive (*Figure 7A*).

We evaluated Tronko using two different alignment methods: Needleman–Wunsch and wavefront alignment. In many cases, the two alignment algorithms perform similarly. However, in the case, where short, single-end reads are used (i.e., 150 bp single-end reads), the wavefront alignment performs worse than the Needleman–Wunsch alignment (see *Figure 2—figure supplement 2D–F* and *Figure 4—figure supplement 2D–F*). The wavefront alignment algorithm implements heuristic modes to accelerate the alignment, which performs similarly to Needleman–Wunsch when the two sequences being aligned are similar in length. However, when there is a large difference between the two sequences being aligned, we notice that the wavefront alignment forces an end-to-end alignment which contains large gaps at the beginning and end of the alignment. Hence, based on current implementations, we cannot recommend the use of the wavefront alignment for assignment purposes of short reads, although this conclusion could change with future improvements in the implementation of the wavefront alignment algorithm.

Tronko is currently not applicable to eukaryotic genomic data as it requires well-curated alignments of markers and associated phylogenetic trees, although we note that whole-genome phylogenetic reference databases for such data could potentially be constructed. Such extensions of the use of Tronko would require heuristics for addressing the memory requirements. Tronko currently has larger memory requirements than methods that are not phylogeny-based. Nonetheless, for assignment to viruses, amplicon sequencing, and other forms of non-genomic barcoding, Tronko provides a substantial improvement over existing assignment methods and is the first full phylogenetic assignment method applicable to modern large datasets generated using NGS.

The methods presented in this article are implemented in the Tronko software package that includes Tronko-build and Tronko-assign for reference database building and species assignment, respectfully. Tronko can be downloaded at https://github.com/lpipes/tronko and is available under an open-software license (*Mapper and Pipes, 2024*).

## Methods
### Tronko-build reference database construction with a single tree

The algorithm used for assignment takes advantage of pre-calculated posterior probabilities of nucleotides at internal nodes of a phylogeny. We first estimate the topology and branch lengths of the tree using RAxML (*Stamatakis, 2014*), although users of the method could use any tree estimation

algorithm. We then calculate and store the posterior probabilities of each nucleotide in each node of the tree. For computational efficiency, this is done under a *Jukes and Cantor, 1969* model, but the method can easily be extended to other models of molecular evolution. The calculations are achieved using an algorithm that traverses the tree only twice to calculate posterior probabilities simultaneously for all nodes in the tree. In brief, fractional likelihoods are first calculated in each node using a standard postorder traversal (e.g., *Felsenstein, 1981*). This directly provides the posterior probabilities in the root after appropriate standardization. An preorder traversal of the tree is then used to pull fractional likelihoods from the root down the tree to calculate posterior probabilities. While naive application of standard algorithms for calculating posterior probabilities in a node, to all nodes of a tree, has computational complexity that is quadratic in the number of nodes, the algorithm used here is linear in the number of nodes as it calculates posterior probabilities for all nodes using a single postorder and a single preorder traversal without having to repeat the calculation for each node in the tree. For a single site, let the fractional likelihood of nucleotide $a \in \{A, C, T, G\}$ in node $j$ be $f_j(a)$, that is, $f_j(a)$ is the probability the observed data in the site for all descendants of node $j$ given nucleotide $a$ in node $j$. Let $h_j(a)$ be the probability of the data in the subtree containing all leaf nodes that are not descendants of node $j$, given nucleotide $a$ in node $j$, then the posterior probability of nucleotide $a$ is (*Yang et al., 1995*)

$$p(a|x) = \frac{f_j(a)h_j(a)\pi_a}{\sum_{b \in \{A,C,G,T\}} f_j(b)h_j(b)\pi_b} \tag{1}$$

where $\pi_a$ is the stationary probability of nucleotide $a$. The algorithm here proceeds by first calculating and storing $f_j(a)$ for all values of $j$ and $a$ using a postorder traversal. It then recursively calculates $h_j(a)$ assuming time-reversibility using a preorder traversal as

$$h_j(a) = \sum_{b \in \{A,C,G,T\}} p_{ab}(t_j)h_{A(j)}(b) \sum_{c \in \{A,C,G,T\}} p_{bc}(t_{S(j)})f_{S(j)}(c) \tag{2}$$

where $t_j$ is the branch length of the edge from node $j$ to its parent, $p_{ab}(t)$ is the time-dependent transition probability of a transition from nucleotide $a$ to nucleotide $b$ in time $t$, $A(j)$ is the parent node of node $j$, and $S(j)$ is the sister node of node $j$ in the binary tree. The algorithm starts at the root with $h_{root}(a) = 1$ This algorithm is implemented in the program 'Tronko-build'.

Each node in the tree is subsequently provided a taxonomy assignment. This is done by first making taxonomic assignments of the leaf nodes using the taxonomy provided by the taxid of the associated NCBI accession. We then make taxonomic assignments for internal nodes, at all taxonomic levels (species, genus, etc.), using a postorder traversal of the tree that assigns a taxonomic descriptor to node $i$ if both children of node $i$ have the same taxonomic assignment. Otherwise, node $i$ does not have a taxonomic assignment at this taxonomic level and node $i$ is given the next closest upward taxonomic level where its children have the same taxonomic assignment. In other words, node $i$ only gets a taxonomic assignment if the taxonomic assignments of both child nodes agree.

## Tronko-build reference database construction with multiple trees

MSAs for a large number of sequences can become unreliable and computationally challenging to work with due to the large number of insertions and deletions. For that reason, we devise an algorithm for partitioning of sequence sets into smaller subsets based on the accuracy of the alignment and using the inferred phylogenetic tree to guide the partitioning (*Figure 1—figure supplement 1*).

To measure the integrity of the MSA, we calculate an average quality score, sum-of-pairs, $ASP$, which is a sum of pairwise alignment scores in the MSA. Assume an MSA of length $l$ with $K$ sequences, $A = \{a_{i,j}\}$, where $a_{i,j}$ is the $j$th nucleotide in sequence $i$, $1 \le i \le K$, $1 \le j \le l$, $a_{i,j} \in M = \{-, A, C, T, G, N\}$. Define the penalty function, $p$:

$$p(I,V) = \begin{cases} 3 & \text{if } I = V \text{ and } I \neq - \text{ (match)} \\ -2 & \text{if } I \neq V, I, V \notin \{N, -\} \text{ (mismatch)} \\ -1 & \text{otherwise} \end{cases} \tag{3}$$

where $I, V \in M$. $ASP$ is then calculated as

$$ASP = \frac{\sum_{j=1}^{l} \sum_{i=1}^{K} \sum_{k=i+1}^{K} p(a_{i,j}, a_{k,j})}{\binom{K}{2}} \tag{4}$$

If the *ASP* is lower than the *ASP* threshold (a threshold of 0.1 was used in our analyses in this article), the corresponding tree is split in three partitions at the node with the minimum variance, calculated as

$$v = \text{argmin}_{i \in T}\left\{((L_1(i) - K/3)^2 + (L_2(i) - K/3)^2) + (K - L_1(i) - L_2(i) - K/3)^2\right\} \tag{5}$$

where $T$ is a tree, that is, a set of nodes, $L_1(i)$ and $L_2(i)$ is the number of leaf nodes descending from the left and right child node, respectively, of node $i$, and $K$ is the total number of leaf nodes in the tree. We then split the tree into three subtrees by eliminating node $v$. Each partition is realigned with FAMSA (*Deorowicz et al., 2016*) and new trees are constructed using RAxML (*Stamatakis, 2014*) using default parameters and the GTR + Gamma model. FAMSA is used to optimize for speed since it is 1–2 orders of magnitude faster than Clustal (*Higgins and Sharp, 1988*) or MAFFT (*Katoh et al., 2002*) with similar quality (see *Deorowicz et al., 2016*). We explored different combinations of tree estimation methods (including IQ-TREE2 *Minh et al., 2020*), multiple sequence aligners, and global aligners (*Figure 2—figure supplement 2*). While most combinations of methods were quite similar (especially for the genus level), the use of FAMSA + RAxML + NW was optimal with regard to speed and accuracy. We ran IQ-TREE2 with the default settings using options `-m GTR+G -nt 4` to be consistent with similar RAxML settings. The sequences are recursively partitioned until the *ASP* score is above the threshold. Finally, the trees, MSAs, taxonomic information, and posterior probabilities are written to one reference file which can be loaded for subsequent assignment of reads. Note that the procedure for phylogeny estimating and calculation of posterior probabilities only has to be done once for a marker and then can be used repeatedly for assignment using different datasets of query sequences.

## Simulation of query sequences

To simulate single-end reads from a reference sequence, a starting point is selected uniformly at random and extends for $m_0$ base pairs, where $m_0$ represents the read length. For paired-end reads, a similar random selection of a starting point occurs, extending $m_0$ base pairs. From the end of this read, if the insert size $m_1$ is positive, the reverse read begins $m_1$ base pairs forward with a length of $m_0$. If $m_1$ is negative, the reverse read starts $m_1$ base pairs backward. Sequencing errors are then added independently with different probabilities $\alpha = 0$, $\alpha = 0.01$, and $\alpha = 0.02$ at each site. These errors are induced by changing the nucleotide to any of the other three possible nucleotides, following the probabilities used by *Stephens et al., 2016*:

$$\begin{array}{c c} & \begin{array}{cccc} A & C & G & T \end{array} \\ \begin{array}{c} A \\ C \\ G \\ T \end{array} & \left[ \begin{array}{cccc} 0 & 0.4918 & 0.3377 & 0.1705 \\ 0.5238 & 0 & 0.2661 & 0.2101 \\ 0.3754 & 0.2355 & 0 & 0.3890 \\ 0.2505 & 0.2552 & 0.4942 & 0 \end{array} \right] \end{array}$$

## Taxonomic classification of query sequences

First, BWA-MEM (*Li, 2013*) is used with default options to align the query sequences to the reference sequences, thereby identifying a list of the highest scoring reference sequences (which we designate as BWA-MEM hits) from the reference database. We use BWA-MEM as the original Minimap2 manuscript (*Li, 2018*) demonstrated that BWA-MEM had the lowest error rate for the same amount of fractional mapped reads compared to Minimap2, SNAP (*Zaharia et al., 2011*), and bowtie2 (*Langdon, 2015*). Second, a global alignment, either using the Needleman–Wunsch algorithm (*Needleman and Wunsch, 1970*) or the wavefront alignment algorithm (*Marco-Sola et al., 2021*), is performed only on the sequence with the highest score from each subtree (reference sequence set) identified using the previously described partitioning algorithm.

Once aligned to the reference sequence, a score, $S(i)$, is calculated for all nodes, $i$, in the tree(s) that the reference sequence is located to. For a given read, let $b_j$ be the observed nucleotide in the position of the read mapping to position $j$ in the alignment. We also assume an error rate, $c$. For

example, if the true base is G and the error rate is $c$, then the probability of observing A in the read is $c/3$. We note that this error rate can be consider to include both true sequencing errors and polymorphisms/sequence divergence. In an ungapped alignment, the score for site $j$ in node $i$ is then the negative log of a function that depends on the posterior probability of the observed nucleotide in the query sequence, $\mathbb{P}_{ij}(b_j)$, and the error rate:

$$-\log(c/3 + (1 - 4c/3)\mathrm{P}_{ij}(b_j)) \tag{6}$$

Assuming symmetric error rates, the probability of observing the base by error is $(1 - \mathbb{P}_{ij}(b_j))c/3$ and the probability of observing the base with no error is $(1 - c)\mathbb{P}_{ij}(b_j)$. The sum of these two expressions equals the expression in the logarithm above. The score for all $s$ sites in the read is defined as $-\sum_{j=1}^{s} \log(c/3 + (1 - 4c/3)\mathbb{P}_{ij}(b_j))$.

Note that the full phylogenetic likelihood for the entire tree, under standard models of molecular evolution (*Yang et al., 1995*) with equal base frequencies and not accounting for errors, and assuming time reversibility, is

$$\ell(t) = \sum_{j=1}^{s} \log\left( \sum_{v \in \{A,C,T,G\}} \mathrm{P}_{ij}(v)p_{vb_j}(t) \right) \tag{7}$$

where $p_{vb_j}(t)$ is the time-dependent transition probability from base $v$ to base $b_j$ in time $t$. This statement takes advantage of the fact that, under time-reversibility, the posterior for a base in an node is proportional to the fractional likelihood of that base in the node, if the tree is rooted in the node. For small values of $t$, $\ell$ converges to $\log(\mathbb{P}_{ij}(b_j))$. Minimizing the score function, therefore, corresponds to maximizing the full phylogenetic likelihood function assuming that the branch leading to the query sequence is infinitesimally short and connects with the tree in an existing node. An alternative interpretation is that the score maximizes the probability of observing the query sequence if it is placed exactly in a node or, equivalently, minimizes the expected mismatch between the query and a predicted sequence sampled form the node.

To address insertions and deletions, we define scores of $\gamma$ and $\lambda$ for a gap or insertion, respectively, in the query sequence relative to the reference sequence. We also entertain the possibility of a gap in the reference sequence in node $i$ in read position $j$, $r_{ij}$, which occurs when the reference is a leaf node with a gap in the position or if it is an internal node with all descendent nodes having gaps in the position. We use the notation $M_g = \{-, N\}$ for gaps and $M_n = \{A, C, T, G\}$ for nucleotides (no gap). Then, the score for node $i$ in site $j$ of the read, with observed base $b_j$, is

$$S_j(i) = \begin{cases} c/3 + (1 - 4c/3)\mathrm{P}_i(b_j) & \text{if } b_j \in M_n \text{ and } r_{ij} \in M_n \\ \gamma & \text{if } b_j \in M_g \text{ and } r_{ij} \in M_n \\ 1 & \text{if } b_j \in M_g \text{ and } r_{ij} \in M_g \\ \lambda & \text{if } b_j \in M_n \text{ and } r_{ij} \in M_g \end{cases} \tag{8}$$

The total score for the entire read is

$$S(i) = \sum_{j=1}^{l} \log(S_j(i)) \tag{9}$$

For paired reads, the scores for each node in the tree are calculated as the sum of the scores for the forward read and the scores for the reverse read. Scores are calculated for all nodes in each tree that contain a best hits from the bwa mem alignment. For all analyses in this article, we use values of $c = 0.01$, $\lambda = 0.01$, and $\gamma = 0.25$.

After calculation of scores, the LCA of all of the lowest scoring nodes, using a user-defined cut-off parameter, is calculated. For example, if the cut-off parameter is 0, only the highest scoring node (or nodes with the same score as the highest scoring node) is used to calculate the LCA. If the cut-off parameter is 5, the highest scoring node, along with all other nodes within a score of 5 of the highest scoring node, are used to calculate the LCA. Once the LCA node is identified, the classification of the

single read (or paired-reads) will be assigned to the taxonomy assigned to that node. The classification of query sequences is parallelized.

## Taxonomic assignment using pplacer

To generate phylogenetic placements using pplacer, we first aligned sequencing reads to the reference sequences using hmmer3 (*Mistry et al., 2013*). We then ran pplacer, rppr prep_db, and guppy classify all using the default parameters in that order. Next, to obtain taxonomic assignments, we used the R package BoSSA (*Lefeuvre, 2018*) to merge the multiclass element (which is a data frame with the taxonomic assignments of each placement) and the placement table of pplace object (the output of pplacer) and only kept the 'best' type of placement for each read. For paired-end sequences, we assigned the taxonomy by the LCA of both pairs of reads.

## Taxonomic assignment using APPLES-2

To generate phylogenetic placements using APPLES-2, we first aligned sequencing reads to the reference sequences using hmmer3 (*Mistry et al., 2013*). We then converted the alignment output from Stockholm to FASTA format and then separated the reference sequences from the sequencing reads (an input requirement for APPLES-2) using in-house scripts. We then ran `run_apples.py` with the default parameters. In order to ensure that the tree that was output from APPLES-2 was strictly binary (a requirement to assign taxonomy), we extracted the tree from the jplace output and resolved polytomies using the `multi2di` function from the R package ape (*Paradis and Schliep, 2019*). Next, we ran `run_apples.py` again using the output tree from ape (with option `--tree=`) and disabled reestimation of branch lengths (in order to keep the tree as strictly binary) by using the option `--disable-reestimation`. To assign taxonomy we ran `gappa examine assign` from the Gappa toolkit (*Czech et al., 2020*) using the options `--per-query-results` and `--best-hit`.

## Classification metrics used for accuracy evaluations

We used the taxonomic identification metrics from *Siegwald et al., 2017* and *Sczyrba et al., 2017*. A true-positive (TP) read at a certain taxonomic rank has the same taxonomy as the sequence it was simulated from. A misclassification (FP) read at a certain taxonomic rank has a taxonomy different from the sequence it was simulated from. A false-negative (FN) read, at a certain taxonomic rank, is defined as a read that received no assignment at that rank. For accuracy, we use the following measures for recall and misclassification rate.

$$\text{Recall} = \frac{TP}{TP + FN} \tag{10}$$

$$\text{Misclassification rate} = \frac{FP}{TP + FP + FN} \tag{11}$$

## Classification of mock community reads

For *Schirmer et al., 2015*, we used the ERR777705 sample, for *Gohl et al., 2016* we used the SRR3163887 sample, and for *Braukmann et al., 2019* we used the SRR8082172 sample. For *Lluch et al., 2015*, we used the ERR1049842 sample. All sample raw reads used for assignment were first filtered through the Anacapa Quality Control pipeline (*Curd et al., 2019*) with default parameters up until before the amplicon sequence variant (ASV) construction step. Only paired reads were retained for assignment. For mock datasets where the true species were only defined with 'sp.', species assignment were excluded for all methods. After Tronko assignments, we filtered results using a script to check the number of mismatches in the forward vs. reverse reads, and used a $\chi^2$ distribution to filter out assignments that have a discrepancy in mismatches.

## Leave-species-out and leave-one-individual-out analyses

We used two datasets (Charadriiformes and Bacteria) for leave-species-out and leave-one-individual-out analyses. For one dataset, we used 1467 COI reference sequences from 253 species from the order Charadriiformes. For the leave-species-out analyses with Charadriiformes, we removed each of the species one at a time (excluding singletons, i.e., species only represented by a single sequence), yielding 252 different reference databases. For the leave-species-out analyses with Bacteria, we randomly selected 5000 taxonomically divergent bacteria species from the 16S reference database

built through CRUX. For the leave-species-out analyses with Bacteria, we removed each of the species, one at a time (excluding singletons), yielding 2323 different reference databases. For each database, we then simulated reads from the species that had been removed with different error rates, and assigned to taxonomy using all methods tested (Tronko, kraken2, metaphlan2, MEGAN, pplacer, and APPLES-2), using the same reference databases and same simulated reads for all methods. For the leave-individual-out analysis with Charadriiformes, we removed a single individual from each species (excluding singletons) yielding 1423 different reference databases. Assignments for all method were performed with default parameters and where a paired read mode was applicable, that mode was used when analyzing paired reads. For paired-end read assignments with MEGAN, the assignment is the LCA of the forward and reverse read assignments as described in the MEGAN manual v6.12.3. For metaphlan, the results from the forward reads and reverse reads were combined.

### Custom 16S and COI Tronko-build reference database construction

For the construction of the reference databases in this article, we use custom-built reference sequences that were generated using common primers (*Caporaso et al., 2012*; *Leray et al., 2013*; *Geller et al., 2013*; *Amaral-Zettler et al., 2009*) for 16S and COI amplicons that have been used in previous studies (*de Vargas et al., 2015*; *Leray and Knowlton, 2015*; *David et al., 2014*) using the CRUX module of the Anacapa Toolkit (*Curd et al., 2019*). For the COI reference database, we use the following forward primer: GGWACWGGWTGAACWGTWTAYCCYCC, and reverse primer: TANACYTCnGGRT-GNCCRAARAAYCA from *Leray et al., 2013* and *Geller et al., 2013*, respectively, as input into the CRUX pipeline (*Curd et al., 2019*) to obtain a fasta and taxonomy file of reference sequences. For the 16S database, we use forward primer GTGCCAGCMGCCGCGGTAA and reverse primer GACTACHV GGGTATCTAATCC from *Caporaso et al., 2012*. We set the length of the minimum amplicon expected to 0 bp, the length of the maximum amplicon expected to 2000 bp, and the maximum number of primer mismatches to 3 (parameters `-s 0`, `-m 2000`, `-e 3`, respectively). Since all of the custom-built libraries contain ≥500,000 reference sequences and MSAs, we first used Ancestralclust (*Pipes and Nielsen, 2022*) to do an initial partition of the data, using parameters of 1000 seed sequences in 30 initial clusters (parameters `-r 1000` and `-b 30`, respectively). For the COI database, we obtain 76 clusters and for the 16S database we obtain 228 clusters. For each cluster, we use FAMSA (*Deorowicz et al., 2016*) with default parameters to construct the MSAs and RAxML (*Stamatakis, 2014*) with the model GTR+$\Gamma$ of nucleotide substitution to obtain the starting trees for Tronko-build.

### Inclusion and diversity

We support inclusive, diverse, and equitable conduct of research.

## Acknowledgements

We thank Rachel Meyer and CALeDNA for their support in this project. We acknowledge Thorfinn Sand Korneliussen for advice on parallelization of the method. This work used the Advanced Cyber-infrastructure Coordination Ecosystem: Services & Support (ACCESS) Bridges system at the Pittsburgh Supercomputing Center through allocation BIO180028 and was supported by NIH grants 1R01GM138634-01 and 1K99GM144747-01.

## Additional information

### Funding

| Funder | Grant reference number | Author |
| --- | --- | --- |
| National Institute of General Medical Sciences | 1R01GM138634-01 | Lenore Pipes Rasmus Nielsen |
| National Institute of General Medical Sciences | 1K99GM144747-01 | Lenore Pipes Rasmus Nielsen |
| Pittsburgh Supercomputing Center | BIO180028 | Lenore Pipes Rasmus Nielsen |

| Funder | Grant reference number | Author |
|---|---|---|

The funders had no role in study design, data collection and interpretation, or the decision to submit the work for publication.

## Author contributions

Lenore Pipes, Conceptualization, Software, Formal analysis, Funding acquisition, Validation, Visualization, Methodology, Writing - original draft, Writing - review and editing; Rasmus Nielsen, Conceptualization, Software, Supervision, Funding acquisition, Methodology, Writing - original draft, Writing - review and editing

## Author ORCIDs

Lenore Pipes ⓘ https://orcid.org/0000-0003-0056-8045

## Decision letter and Author response

Decision letter https://doi.org/10.7554/eLife.85794.sa1
Author response https://doi.org/10.7554/eLife.85794.sa2

---

# Additional files

## Supplementary files

• MDAR checklist

## Data availability

The identified reference databases, MSAs, phylogenetic trees, and posterior probabilities of nucleotides in nodes for COI and 16S are available for download at https://doi.org/10.5281/zenodo. 13182507.

The following dataset was generated:

| Author(s) | Year | Dataset title | Dataset URL | Database and Identifier |
|---|---|---|---|---|
| Pipes L, Nielsen R | 2024 | A rapid phylogeny-based method for accurate community profiling of large-scale metabaroding datasets | https://doi.org/ 10.5281/zenodo. 13182507 | Zenodo, 10.5281/ zenodo.13182507 |

The following previously published datasets were used:

| Author(s) | Year | Dataset title | Dataset URL | Database and Identifier |
|---|---|---|---|---|
| Schirmer M, Ijaz UZ, D'Amore R, Hall N, Sloan WT, Quince C | 2015 | Mock community sequencing | https://trace.ncbi. nlm.nih.gov/Traces/? view=run_browser& acc=ERR777705& display=metadata | NCBI Sequence Read Archive, ERR777705 |
| Gohl DM, Vangay P, Garbe J, MacLean A, Hauge A, Becker A, Gould TJ, Clayton JB, Johnson TJ, Hunter R, Knights D, Beckman KB | 2016 | 16S Methods Comparison, raw sequence reads | https://trace.ncbi. nlm.nih.gov/Traces/? view=run_browser& acc=SRR3163887& display=metadata | NCBI Sequence Read Archive, SRR3163887 |

*Continued*

| Author(s) | Year | Dataset title | Dataset URL | Database and Identifier |
|---|---|---|---|---|
| Braukmann TWA, Ivanova NV, Prosser SWJ, Elbrecht V, Steinke D, Ratnasingham S, de Waard JR, Sones JE, Zakharov EV, Hebert PDN | 2019 | Revealing the Complexities of Metabarcoding with a Diverse Arthropod Mock Community | https://trace.ncbi.nlm.nih.gov/Traces/?view=run_browser&acc=SRR8082172&display=metadata | NCBI Sequence Read Archive, SRR8082172 |
| Lluch J, Servant F, Paisse S, Valle C, Valiere S, Kuchly C, Vilchez G, Donnadieu C, Courtney M, Burcelin R | 2015 | The characterization of novel tissue microbiota using an optimized 16S metagenomic sequencing pipeline | https://trace.ncbi.nlm.nih.gov/Traces/?view=run_browser&acc=ERR1049842&display=metadata | NCBI Sequence Read Archive, ERR1049842 |

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
