## [Editor Report]

This important work presents a novel tool for performing phylogenetic assignment of DNA sequences. The manuscript is convincing, and the authors perform a standard benchmark experiment against current state-of-the-art tools using real + simulated datasets to demonstrate where the novel tool stands in the context of existing methods. This paper will be of great interest to bioinformaticians and evolutionary biologists interested in massively-scalable phylogenetic assignment.

---

## [Decision Letter]

**Decision letter after peer review:**

Thank you for submitting your article "A rapid phylogeny-based method for accurate community profiling of large-scale metabarcoding datasets" for consideration by *eLife*. Your article has been reviewed by 3 peer reviewers, including Niema Moshiri as the Reviewing Editor and Reviewer #1, and the evaluation has been overseen by Wendy Garrett as the Senior Editor. The following individual involved in review of your submission has agreed to reveal their identity: Nick Goldman (Reviewer #3).

Essential revisions:

1) There were some concerns that the data sets that were chosen in the manuscript might not be well-aligned with real-world use cases for the tool, and that other data sets could be included that better represent real-world use cases; see Reviews below for specific comments. The realism of the chosen data sets should be justified, and/or datasets more representative of real-world scenarios could be included.

2) The manuscript includes some comparisons between Tronko and pplacer, and it excludes pplacer from the benchmark experiment due to pplacer's inability to scale to the necessary dataset sizes (which is perfectly reasonable). However, a recently published tool named APPLES (Balaban et al., 2020; https://doi.org/10.1093/sysbio/syz063) seems to be able to perform taxonomic classification using phylogenetic placement in a similar manner as pplacer (i.e., by feeding the output phylogenetic placement file to guppy to output taxonomic classifications). The following tutorial by the authors of APPLES seems to present this "APPLES to guppy" workflow:

https://github.com/smirarab/tutorials/blob/master/Skmer-APPLES-tutorial.md#run-actual-placement

The Reviewers were unsure about whether or not APPLES (as a potential substitute for pplacer) would be a reasonable alternative for Tronko. If APPLES is a reasonable alternative for Tronko, we request that it be included in any existing benchmarks in which Tronko was compared against pplacer. If APPLES is not a reasonable alternative for Tronko (and should thus not be included in the existing comparison), we request additional details in the introduction of the manuscript describing why APPLES is inappropriate for this task.

3) Some choices were made without clear justification (e.g. the use of specific dependency tools as well as parameter selection for those tools), and these choices require some form of justification and/or discussion/exploration; see Reviews below for specific comments.

4) There were technical issues/errors with the distribution of the tool that need to be fixed; see Reviews below for specific errors/scenarios as well as some potential suggestions.

5) There are minor issues with the writing of the manuscript itself that should be updated/corrected; see Reviews below for specific comments/suggestions.

*Reviewer #1 (Recommendations for the authors):*

First, some technical questions about the methodology:

How is the accuracy impacted by potential errors in the multiple sequence alignment (MSA) and phylogenetic inference procedure? For example, what if someone were to use e.g. MAFFT (Katoh et al., 2002) for MSA followed by FastTree 2 (Price et al., 2010) or IQ-TREE (Nguyen et al., 2015) or matOptimize (Ye et al., 2022) for phylogenetic inference instead of the Tronko-build approach? Or perhaps existing joint MSA + tree inference tools like PASTA (Mirarab et al., 2015)?

Why were existing phylogenetic placement tools excluded, such as UShER (Turakhia et al., 2021), APPLES (Balaban et al., 2019), SEPP (Mirarab et al., 2012), TIPP (Nguyen et al., 2014), or UPP (Nguyen et al., 2015)? The authors exclude pplacer due to runtime + memory intensity, but they did not provide rationale for providing the many other existing phylogenetic placement methods. My understanding is that UShER is supposed to be *extremely* fast and quite memory-efficient.

Tronko currently supports two BWA-MEM modes (Needleman-Wunsch and Wavefront Alignment), but rather than just supporting BWA-MEM, what about other potential aligners? For example, if one were to use Minimap2 to perform alignment instead, how would the results + runtime + memory requirements change?

The manuscript explains that the high memory usage is because of all of the things Tronko needs to store in memory the entire time, but (1) do all of those things really need to be stored in memory simultaneously, and (2) could some form of compressive encoding (e.g. 2-bit encoding for reference genomes) be used to reduce memory usage? 50 GB is reasonable for high-end modern servers, but I think it can be dramatically reduced with clever optimizations. I think the discussion of the peak memory requirements would benefit from more thorough exploration of what exactly is contributing to the large memory consumption (e.g. what proportion of it is from storing the trees, or the reference genomes, or the posterior probabilities, etc.).

Now, general comments about the paper:

The paper is generally well-written and reads fairly clearly, but my main concern is about some choices in the methodology that were (seemingly) somewhat arbitrarily chosen without providing justification. For example, BWA-MEM was chosen for mapping; why that choice rather than other mappers? And why are the default parameters appropriate? RAxML was chosen for phylogenetic inference; why that choice rather than other tree inference tools? And why default parameters with GTR+Γ model (rather than GTR+Γ+Invariant, or GTR+CAT, or GTR+CAT+Invariant, etc.)? In addition to my technical questions above about how changing these choices would impact results, my general comment here is that all choices should be motivated in some way within the text.

Both panels of Figure 6 should be log-scale: as they're currently presented in linear-scale, it's impossible to meaningfully discern differences between the smaller lines.

*Reviewer #2 (Recommendations for the authors):*

## Overall comments

The manuscript is hard to read top-to-bottom, and would be easier if you gave a little more of an overview of the method before giving results. The figure 1 caption, for example, can't really be understood without getting to P14. Figure 1 itself can't really be understood without understanding the cutoff parameter.

Please make it clear from the get-go that this is amplicon sequencing and not true metagenomics. The tool is compared to Kraken which does true metagenomics.

If I am understanding correctly, the database input for the method requires a tree annotated with taxonomic labels at all nodes of the tree. If that's right, what is the process for doing this labeling? What if the taxonomy and the tree disagree?

The paper doesn't compare to APPLES (reference 8) which is phylogenetic but distance-based and therefore should be faster, but is probably less accurate. This isn't necessary for a revision, but it would be interesting to understand if performance gaps are due to phylogeny or likelihood-based phylogeny.

## Details

Please number lines. This makes reviewing much easier.

P2 "There are three…": I suggest using (1), (2), (3) in this sentence to make it clearer rather than just rely on commas.

P3 Tandy Warnow has worked on a number of approaches to scaling likelihood-based phylogenetic placement, e.g. https://www.biorxiv.org/content/10.1101/2022.10.26.513936v1.full.pdf which also uses a collection of trees.

P7 It seems like you are using a different cutoff for the different comparisons. Can you justify that?

P12 "lenghts" typo

P12 The two-pass algorithm is linear time, and this seems to me like the standard two-pass algorithm. We can get the required posterior probabilities from the upward and downward partial likelihood vectors. What am I missing? If there is something subtle, some more explanation and notation is needed.

P12 To get an assignment to node i, does i and its children need to get a given taxonomic assignment? The text reads as if only the children need taxonomic assignments.

P13 Why I and V? I know they are arbitrary but it seems like an odd choice.

P13 What is the intuition for the definition of variance? If we were to try to write this as the variance of a random variable, what would it be? It seems like this definition is similar to that for the centroid of the tree. If that's the case, why not use that more common object?

P14 Perhaps "written" rather than "printed"? Is there a specific format to this file, like JSON, or is it fully custom?

P15 Notation would be cleaner if you just used a single P for posterior probability, or \mathbb{P}. use "\log". Suggest display equation for phylogenetic likelihood.

P16 Interesting that this classifies N as a gap. Sometimes Ns are ambiguous sequencing calls.

P16 What is m(1)?

Figure 2: I found "error/polymorphism" confusing, as at first I thought it was the ratio of the two. Suggest just "error" in the figure and caption, and note in the text that it could be in fact polymorphism. Also, there are lots of triangles for Tronko. Please introduce in the text what the cut-off is. You state that there is a cut-off on P4, and hopefully in one additional sentence you could give an intuition. Font is too small.

Figure 3: These "rainbow" schemes are now not in favor compared to color schemes like viridis and variants; see https://journals.plos.org/plosone/article?id=10.1371/journal.pone.0199239

Figure 6: Please consider color scheme, the yellow is basically invisible.

Figure S4. I'm not sure what "assignment rate" is. Is the fraction of time the program produced any result? Could you label the Tronko lines with Tronko?

All Figure legends should identify the gene target and data set.

*Reviewer #3 (Recommendations for the authors):*

We have divided most of our considerations into three areas, regarding (a) the methods that this ms. introduces, (b) the results that they produce and (c) the Tronko software.

(a) The methods take familiar ground as their starting point, and then take a new direction in terms of approximations and shortcuts to implement something resembling a full probabilistic (maximum likelihood) inference technique, but fast enough and with low-enough memory requirement to be practical for modern-sized datasets (reference and sample). We need no convincing that this is a valid approach, and is sufficient for the methods proposed to be of value if they produce good-enough results.

That said, there were a few instances of other relevant work not being quoted that I think would be of interest to some readers. We don't think this ms. needs to address all these points in a detailed manner, but where other relevant work has not been cited then readers can reasonably ask whether Pipes and Nielsen are unaware or have simply decided not to incorporate some other existing ideas; or whether they tried to incorporate ideas that ultimately did not work; whether some alternative concepts are somehow incompatible with the approach they did take; or whether there is some other reason. Some specific examples we are curious about and would like to see discussed in this manuscript:

* what about methods that combine composition-based (kmer) and phylogenetic approaches, essentially replacing likelihood calculations with kmer statistics in a phylogenetic framework? See work like https://academic.oup.com/bioinformatics/article/35/18/3303/5303992 and others from that group, including recent work by Romashchenko

* since the ms. argues so strongly for the benefits of phylogenetic methods, it would be interesting to see how the authors consider Tronko to compare to phylogeny-based methods other than pplacer, for example papara and PAGAN and others e.g. mentioned in https://www.frontiersin.org/articles/10.3389/fbinf.2022.871393/full

(b) The results seem convincing and reasonable metrics were used for evaluation. However,

* since the method's highlight is incorporating phylogeny, it would be interesting to see some comparisons of results using a metric that takes phylogeny into account. For example, while calculating accuracy of taxonomic assignment, instead of using a binary score (correct/incorrect), could you calculate a distance between assignment and real value? A measure of how wrong incorrect assignments could be of relevance to many users in terms of biological interpretations. Arguably, a tool that misassigns to close species would be preferable over another that misassigns to further species

* very poor performances of some tools in mock community analyses might make readers ask if the default parameters of the tools were right or not for this comparison. For example, for Braukmann et al. data the authors say "MEGAN did not assign any reads at the species or genus level"

* some claims about other methods' results being dominated by Tronko based on the convex hull seem a bit overstated, when there has been no investigation of how the other methods' recall/misclassification response curves could appear if other parameter settings were used

(c) Tronko software is publically available on GitHub with clear documentation, example files, and working example commands. Nevertheless there is room for improvement for Tronko to be widely utilised by the community:

* The software installation and execution can be tested on a few different computers and ideally operating systems. We failed to install the software from the source but Singularity installation works fine.

* tronko-build input files are preprocessed with multiple softwares. Although the authors provide example files, there is not a guide on how to create those files for our own gene of interest. A couple of sentences (or maybe more, for users who are not so familiar with the data types and analyses involved) about how those files were prepared would be very helpful.

* tronko-build example files include some files with double extensions (e.g..fasta.ann). These are not mentioned in the readme. It is not clear what they are. They are not mentioned in the inputs of the command nor did they appear in the output directory when we ran it.

---

## [Author Response]

Essential revisions:1) There were some concerns that the data sets that were chosen in the manuscript might not be well-aligned with real-world use cases for the tool, and that other data sets could be included that better represent real-world use cases; see Reviews below for specific comments. The realism of the chosen data sets should be justified, and/or datasets more representative of real-world scenarios could be included.

By ”real-world scenarios,” we interpreted this feedback to mean the inclusion of more heterogeneous bacterial data sets. In response, we have added more bacteria datasets including 9 missing out tests (Figures 5, S5, and S6) and an additional mock community dataset from Lluch et al. (2015) (Figures 6B and S9). For the missing out tests we include this description on how that dataset was made:

”In order to replicate real-world scenarios, we added a leave-one-species-out test using 16S from 2,323 bacterial species and 5,000 individual sequences (Figure 5). We selected the sequences for the 16S dataset by grouping the sequences by the class level in a random order, rotating that order, and randomly selecting an individual sequence from each group.”

Lluch et al. (2015) designed a mock community that was prepared by cloning the complete 16S rDNA gene of 14 different bacterial species. This inclusions aim to better represent practical applications and enhance the relevance of our method to real-world use cases.

2) The manuscript includes some comparisons between Tronko and pplacer, and it excludes pplacer from the benchmark experiment due to pplacer's inability to scale to the necessary dataset sizes (which is perfectly reasonable). However, a recently published tool named APPLES (Balaban et al., 2020; https://doi.org/10.1093/sysbio/syz063) seems to be able to perform taxonomic classification using phylogenetic placement in a similar manner as pplacer (i.e., by feeding the output phylogenetic placement file to guppy to output taxonomic classifications). The following tutorial by the authors of APPLES seems to present this "APPLES to guppy" workflow:https://github.com/smirarab/tutorials/blob/master/Skmer-APPLES-tutorial.md#run-actual-placementThe Reviewers were unsure about whether or not APPLES (as a potential substitute for pplacer) would be a reasonable alternative for Tronko. If APPLES is a reasonable alternative for Tronko, we request that it be included in any existing benchmarks in which Tronko was compared against pplacer. If APPLES is not a reasonable alternative for Tronko (and should thus not be included in the existing comparison), we request additional details in the introduction of the manuscript describing why APPLES is inappropriate for this task.

In the revised version of the manuscript, we compare against pplacer and APPLES-2 for all datasets where it was feasible. We were not able to include these methods for comparisons in the mock community datasets for the reason that we were unable to construct a multiple sequence alignment that included the entire barcode’s database and the sequencing reads. We attempted to make a multiple sequence alignment for the COI barcode using PASTA (Mirarab et al., 2014) but we abandoned this attempt after the job ran for more than 2 weeks.

Tronko is able to construct a reference database for large databases (i.e., COI reference sequences) because it does not require aligned sequences. We use a method (AncestralClust) that we developed previously to first cluster the sequences, and we align sequences in each cluster. Tronko can take as input a series of trees rather than just one large tree. We describe exactly how these trees were constructed in the Methods in the section ’Custom 16S and COITronko-build reference database construction’.

We added a description of assigning taxonomy using pplacer in the Methods under ”Taxonomic assignment using pplacer”:

”To generate phylogenetic placements using pplacer, we first aligned sequencing reads to the reference sequences using hmmer. We then ran pplacer, rppr prep db, and guppy classify all using the default parameters in that order. Next, we used the R package BoSSA to merge the multiclass (which is a a data frame with the taxonomic assignments of each placement) and the placement table of pplace object (the output of pplacer) and only kept the ”best” type of placement for each read. For paired-end sequences, we assigned the taxonomy by the LCA of both pairs of reads.”

Additionally, we added a description of assigning taxonomy using APPLES-2 in the Methods under ”Taxonomic assignment using APPLES-2”:

”To generate phylogenetic placements using APPLES-2. We first aligned sequencing reads to the reference sequences using hmmer. We then converted the alignment output from Stockholm to FASTA format and then separated the reference sequences from the sequencing reads (an input requirement for APPLES-2) using in-house scripts. We then ran run_apples.py with the default parameters. In order to ensure that the tree that was output from APPLES-2 was strictly binary (a requirement to assign taxonomy), we extracted the tree from the jplace output and resolved polytomies using the multi2di function from the R package ape. Next, we ran run apples.py again using the output tree from ape (with option –tree=) and disabled reestimation of branch lengths (in order to keep the tree as strictly binary) by using the option –disable reestimation. To assign taxonomy we ran gappa examine assign from the Gappa toolkit using the options –per-query- results and –best-hit.”

We ended up having to run APPLES-2 twice (one time to obtain the tree that had consistent placements and the next time to have the placements on a strictly binary tree) because of the issue with the resulting polytomies. We reached out to the first author of ”Environmental DNA metabarcoding of Danish soil samples reveals new insight into the hidden diversity of eutardigrades in Denmark”, Frida Pust, to understand how they were able to obtain taxonomic assignments from APPLES-2. They stated ”I did the taxonomic assignment manually. When my tree had polytomies, I did not trust the placement and simple assigned to genus-level or above – manually. I only had around 100 sequences, so it was not too bad.”

In addition to adding pplacer and APPLES-2 comparisons whenever possible, we added a comparison of APPLES-2 in terms of its speed and memory in Figure S7.

When we included pplacer and APPLES-2 comparisons, we noticed that these methods perform worse than kraken2 for species level assignments (using a leave-one-individual-out test). As we looked closer at this unexpected issue, we have determined that it is due to choosing the placement in the tree with the highest likelihood, when there are many places in the tree with a similar likelihood. Therefore, the highest likelihood is random because of the optimization. Currently, there is no method that combines the likelihoods, realizes that they are the same, and calculates the LCA. We removed pplacer and APPLES-2 from the main figure (Figure 4) because of this issue of poor species level assignment we only show these results in Supplementary Figure S5. Additionally, we added this to the text:

”We believe it would be slightly misleading to display results for pplacer and APPLES-2 here due to the lack of an implementation to calculate the LCA on similar likelihoods.”

Similarly, with Figure 5G-I, we do not display APPLES-2 because of its poor performance, which we deem as misleading. The results for Figure 5 (paired-end, single-end 150bp, and 300bp) are not directly comparable. For this analysis using 300bp leave-one-species-out test, only 105 out of 2,310, number of species had a reference sequence that was >=300bp in length. For reference sequences <300bp, we could not perform leave-one-species-out. We added the following text:

”The simulations for 300bp single-end reads are not directly comparable to the 150bp single-end or paired-end reads because only 105 missing-out tests out of 2,310 were able to be performed because most reference sequence were <300bp in length. We only display the results for 300bp single-end reads for APPLES-2 in the supplement, as we believe the results are not a good representation of the method. See Figure S6 for results for APPLES-2 using 300bp single-end reads along with results using the Wavefront alignment algorithm.”

3) Some choices were made without clear justification (e.g. the use of specific dependency tools as well as parameter selection for those tools), and these choices require some form of justification and/or discussion/exploration; see Reviews below for specific comments.

For tronko-build we use FAMSA and RAxML. Our justification for using FAMSA over other multiple sequence aligners is its speed. We added this text:

”FAMSA is used to optimize for speed since it is 1 to 2 orders of magnitude faster than Clustal or MAFFT with similar quality (see Deorowicz et al., 2016).”

We use FAMSA with its default parameters. As for the use of RAxML. We use this as the standard model in phylogenetics we used GTR+Γ and it is beyond the scope of the manuscript to explore general model fit. As the standard model in phylogenetics we used GTR+Γ but we realized that other models could also have been explored. And perhaps with different datasets what leads to the best phylogenetics accuracy depends on the amount of data and the particular details of the molecular evolution of the markers used and species. While the GTR+Γ models has proven to be identifiable, the original proofs of identifiably for the GTR+Γ+I model have been shown to be invalid and no such proof has been provided to date (Allman et al., 2016). We therefore choose the GTR+Γ model rather than the GTR+Γ+I model.

For tronko-assign, we use two alignment steps. At the first step, we use a short read mapper, bwa-mem, to find the best hit. In Heng Li’s ”Minimap2: pairwise alignment for nucleotide sequences”, he performs a comparison of Minimap2, bwamem, bowtie2, and snap. As demonstrated in Figure 1b of the manuscript, bwa-mem has the lowest error rate for the same amount of fractional mapped reads. The manuscript also states ”Minimap2 is more accurate on this simulated dataset than Bowtie2 and SNAP but less accurate than BWA-MEM”. We added this text:

”We use BWA-MEM as the original Minimap2 manuscript demonstrated that BWA-MEM had the lowest error rate for the same amount of fractional mapped reads when compared to Minimap2, SNAP, and bowtie2.”

Generally, which aligner to use is an interesting problem. As aligners improve in the future that will continue to be an active research problem but it is beyond the scope of this manuscript.

4) There were technical issues/errors with the distribution of the tool that need to be fixed; see Reviews below for specific errors/scenarios as well as some potential suggestions.

We address this in our responses to the specific comments below.

5) There are minor issues with the writing of the manuscript itself that should be updated/corrected; see Reviews below for specific comments/suggestions.

We address this in our responses to the specific comments below.

Reviewer #1 (Recommendations for the authors):First, some technical questions about the methodology:How is the accuracy impacted by potential errors in the multiple sequence alignment (MSA) and phylogenetic inference procedure? For example, what if someone were to use e.g. MAFFT (Katoh et al., 2002) for MSA followed by FastTree 2 (Price et al., 2010) or IQ-TREE (Nguyen et al., 2015) or matOptimize (Ye et al., 2022) for phylogenetic inference instead of the Tronko-build approach?

The reason for not using FastTree2 is that our taxonomic assignment algorithm is not designed to work on trees that contain polytomies. The reason for not using mat Optimize is that it is a maximum parsimony based tree optimization method. Maximum likelihood provides more accurate and reliable phylogenetic trees by incorporating detailed evolutionary models and probabilistic frameworks. IQ-TREE2 is considerably slower than RAxML. On just one of our missing out tests, IQ-TREE2 took 37 minutes and RAxML took just 11 minutes when running on a single thread.

Nevertheless, we have included an analysis Figure S3 where we tested different combinations of aligners and tree estimation methods. In this analysis, the impact of the choice of method was minimal.

Or perhaps existing joint MSA + tree inference tools like PASTA (Mirarab et al., 2015)?

As we previously mentioned, we attempted to use PASTA on our CO1 reference database of >1.5 million sequences, but this job did not finish in any meaningful time.

Why were existing phylogenetic placement tools excluded, such as UShER (Turakhia et al., 2021), APPLES (Balaban et al., 2019), SEPP (Mirarab et al., 2012), TIPP (Nguyen et al., 2014), or UPP (Nguyen et al., 2015)? The authors exclude pplacer due to runtime + memory intensity, but they did not provide rationale for providing the many other existing phylogenetic placement methods. My understanding is that UShER is supposed to be *extremely* fast and quite memory-efficient.

UShER, unlike other placement methods, requires a VCF file as input. VCF format is designed to report variations relative to a single reference sequence (not many reference sequences like as in the case of a metabarcode). We now include APPLES-2 and pplacer whenever possible for all datasets examined. SEPP runs pplacer internally and thus has the same speed and memory limitations of pplacer. TIPP also has limitations in its ability to scale with reference sequences. It uses SATe which Liu et al. (2012) demonstrated that SATe-II took approximately 1 week to finish an iteration of the alignment algorithm for 27,643 sequences. Additionally, UPP is a multiple sequence aligner that is used in SEPP. For comparisons with IQ-TREE2 we added Figure S3 and the following text:

We explored different combinations of tree estimation methods (including IQ-TREE2), multiple sequence aligners, and global aligners (Figure S3). While most combinations of methods were quite similar (especially for the genus level), the use of FAMSA+RAxML+NW was optimal with regards to speed and accuracy. We ran IQ-TREE2 with the default settings using options -m GTR+G -nt 4 to be consistent with similar RAxML settings.

Tronko currently supports two BWA-MEM modes (Needleman-Wunsch and Wavefront Alignment), but rather than just supporting BWA-MEM, what about other potential aligners? For example, if one were to use Minimap2 to perform alignment instead, how would the results + runtime + memory requirements change?

Needleman-Wunsch and Wavefront Alignment are not modes of BWA-MEM. Please see the earlier response to ’Essential Revisions’ #3.

The manuscript explains that the high memory usage is because of all of the things Tronko needs to store in memory the entire time, but (1) do all of those things really need to be stored in memory simultaneously, and (2) could some form of compressive encoding (e.g. 2-bit encoding for reference genomes) be used to reduce memory usage? 50 GB is reasonable for high-end modern servers, but I think it can be dramatically reduced with clever optimizations. I think the discussion of the peak memory requirements would benefit from more thorough exploration of what exactly is contributing to the large memory consumption (e.g. what proportion of it is from storing the trees, or the reference genomes, or the posterior probabilities, etc.).

It is possible to store one tree at a time, which would greatly reduce that memory consumption of a large reference database such as COI, however, this would increase the run time as the memory for one tree (including its posterior probabilities) would need to be unloaded and then loaded into memory again. The vast majority of the memory consumption comes from having to store posterior probabilities. However, it is possible to store these probabilities using a tree compression algorithm from Larget & Simon (1998) that may be implemented in a future version.

Now, general comments about the paper:The paper is generally well-written and reads fairly clearly, but my main concern is about some choices in the methodology that were (seemingly) somewhat arbitrarily chosen without providing justification. For example, BWA-MEM was chosen for mapping; why that choice rather than other mappers? And why are the default parameters appropriate? RAxML was chosen for phylogenetic inference; why that choice rather than other tree inference tools? And why default parameters with GTR+Γ model (rather than GTR+Γ+Invariant, or GTR+CAT, or GTR+CAT+Invariant, etc.)? In addition to my technical questions above about how changing these choices would impact results, my general comment here is that all choices should be motivated in some way within the text.

We agree with the reviewer that all choices should be motivated. We explain in our response to ’Essential Revisions’ #3 the answers to these specific concerns.

Both panels of Figure 6 should be log-scale: as they're currently presented in linear-scale, it's impossible to meaningfully discern differences between the smaller lines.

This has been modified in the new draft of the manuscript.

Reviewer #2 (Recommendations for the authors):## Overall commentsThe manuscript is hard to read top-to-bottom, and would be easier if you gave a little more of an overview of the method before giving results. The figure 1 caption, for example, can't really be understood without getting to P14. Figure 1 itself can't really be understood without understanding the cutoff parameter.Please make it clear from the get-go that this is amplicon sequencing and not true metagenomics. The tool is compared to Kraken which does true metagenomics.

The title of the manuscript is ’A rapid phylogeny-based method for accurate community profiling of large-scale metabarcoding datasets’.

If I am understanding correctly, the database input for the method requires a tree annotated with taxonomic labels at all nodes of the tree. If that's right, what is the process for doing this labeling? What if the taxonomy and the tree disagree?

Actually, this is not correct. The method only requires taxonomic labels for the leaf nodes of the tree. These taxonomic labels are gathered from the NCBI taxonomy database. We describe the algorithm in detail in the Methods section under ’Tronko-build reference database construction with a single tree’ in lines 257 to 263.

The paper doesn't compare to APPLES (reference 8) which is phylogenetic but distance-based and therefore should be faster, but is probably less accurate. This isn't necessary for a revision, but it would be interesting to understand if performance gaps are due to phylogeny or likelihood-based phylogeny.

We added comparisons to APPLES-2 to all applicable datasets. We described this in detail with our response to Essential Revision #2.

## DetailsPlease number lines. This makes reviewing much easier.

This was added.

P2 "There are three…": I suggest using (1), (2), (3) in this sentence to make it clearer rather than just rely on commas.

This was added.

P3 Tandy Warnow has worked on a number of approaches to scaling likelihood-based phylogenetic placement, e.g. https://www.biorxiv.org/content/10.1101/2022.10.26.513936v1.full.pdf which also uses a collection of trees.

When we say that ”phylogeny-based method shave not scaled to handle the entirety of the rapidly growing reference databases of genome markers and the increasingly large amounts of NGS data”, we mean that these methods have not scaled to be able to simultaneously handle millions of reference sequences and millions of queries. Although BATCH-SCAMPP (Wedell et al., 2022) uses subtrees, it still requires as input 1 phylogenetic tree along with a multiple sequence alignment of both reference sequences and query sequences. In addition to this, there are a number of limiting factors in BATCH-SCAMMP. In Wedell et al. (2022), they recommend to limit the subtree size to 2,000 reference sequences and the largest backbone size they tested only had 50,000 reference sequences.

P7 It seems like you are using a different cutoff for the different comparisons. Can you justify that?

In all analyses we compared all methods to Tronko using 5 cut-offs 0, 5, 10, 15, and 20. In any case, we amended the text to compare with the other methods using a cut-off of 10:

”Using 150bp paired-end reads with 0% error (Figure 4D), Tronko had a misclassification rate at only 0.1% with a recall rate of 58.6% at the species level using a cut-off set to 10 while kraken2, MEGAN, and metaphlan2 had misclassification rates of 1.5%, 0.1%, and 11.0%, respectively, with recall rates of 85.4%, 60.7%, and 98.14%. ”

P12 "lenghts" typo

We amended: ”We first estimate the topology and branch-lengths of the tree using RAxML, although users of the method could use any tree estimation algorithm.”

To read: ”We first estimate the topology and branch-lengths of the tree using RAxML, although users of the method could use any tree estimation algorithm.”

P12 The two-pass algorithm is linear time, and this seems to me like the standard two-pass algorithm. We can get the required posterior probabilities from the upward and downward partial likelihood vectors. What am I missing? If there is something subtle, some more explanation and notation is needed.

Standard implementations of the two-pass algorithm requires this algorithm to be run for each node of the tree when posterior probabilities need to be calculated for all nodes. Our algorithm allows calculations to be done for all nodes using a single postorder traversal and a single inorder traversal. We have now added the following to the manuscript:

”…, as it calculates posterior probabilities for all nodes using a single postorder and inorder traversal without having to repeat the calculation for each node in the tree”.

As this is a fairly obvious idea, we are thinking that there might be other implementations of this, but we could not find references in the literature to this. If we have missed this and the reviewer is aware of a reference to this, we are happy to add this reference instead of our description.

P12 To get an assignment to node i, does i and its children need to get a given taxonomic assignment? The text reads as if only the children need taxonomic assignments.

Yes, *i* and its children need to have a taxonomic assignment. To clarify the algorithm further we changed:

”We then make taxonomic assignments for internal nodes, at all taxonomic levels (species, genus, etc), using a postorder traversal of the tree that assigns a taxonomic descriptor to node *i* if both children of node *i* have the same taxonomic assignment. Otherwise, node *i* does not have a taxonomic assignment at this taxonomic level. In other words, node *i* only gets a taxonomic assignment if the taxonomic assignments of both child nodes agree.”

To read:

”We then make taxonomic assignments for internal nodes, at all taxonomic levels (species, genus, &c.), using a postorder traversal of the tree that assigns a taxonomic descriptor to node *i* if both children of node *i* have the same taxonomic assignment. Otherwise, node *i* does not have a taxonomic assignment at this taxonomic level and node *i* is given the next closest upwards taxonomic level where its children have the same taxonomic assignment.”

P13 Why I and V? I know they are arbitrary but it seems like an odd choice.

We are running out of letters for notation and we would like all notations to be distinct.

P13 What is the intuition for the definition of variance? If we were to try to write this as the variance of a random variable, what would it be? It seems like this definition is similar to that for the centroid of the tree. If that's the case, why not use that more common object?

A centroid is not necessarily unique. One tree might have multiple centroids. Using the variance is a natural way of identifying partitions of a tree of approximately equal size using an L2 penalty on large clusters.

P14 Perhaps "written" rather than "printed"? Is there a specific format to this file, like JSON, or is it fully custom?

It is a fully custom format specific to Tronko. We have changed:

”Finally, the trees, multiple sequence alignments, taxonomic information, and posterior probabilities are printed to one reference file which can be loaded for subsequent assignment of reads.”

To read:

”Finally, the trees, multiple sequence alignments, taxonomic information, and posterior probabilities are written to one reference file which can be loaded for subsequent assignment of reads.”

P15 Notation would be cleaner if you just used a single P for posterior probability, or P. use "log". Suggest display equation for phylogenetic likelihood.

We amended this:

”In an ungapped alignment, the score for site *j* in node *i* is then the negative log of a function that depends on the posterior probability of the observed nucleotide in the query sequence, *PP_ij_*(*b_j_*), and the error rate:−log⁡(c/3 +(1−4c/3)PPij(bj))

Assuming symmetric error rates, the probability of observing the base by error is (1 − *PP_ij_*(*b_j_*))*c*/3 and the probability of observing the base with no error is (1 − *c*) *PP_ij_*(*b_j_*). The sum of these two expressions equals the expression in the logarithm above. The score for all *s* sites in the read is defined as −∑j=1slog⁡(c3+(1−4c/3)PPij(bj))

Notice that the full phylogenetic likelihood for the entire tree, under standard models of molecular evolution with equal base frequencies and not accounting for errors is ℓ=∑j=1slog⁡(∑v∈{A,C,T,G}PPij(v)Pvbj(t)), where *P_vbj_*(*t*) is the time dependent transition probability from base *v* to base *b_j_* in time *t*. This statement takes advantage of the fact that, under time-reversibility, the posterior for a base in an node is proportional to the fractional likelihood of that base in the node, if the tree is rooted in the node. For small values of *t*, ℓ converges to log(*PP_ij_*(*b_j_*)).”

To read:

”In an ungapped alignment, the score for site *j* in node *i* is then the negative log of a function that depends on the posterior probability of the observed nucleotide in the query sequence, *IP^ij^*(*b_j_*), and the error rate:−log(c/3+(1−4c/3)]IPij(bj)

Assuming symmetric error rates, the probability of observing the base by error is (1 – *IP^ij^*(*b_j_*)*c*/3) and the probability of observing the base with no error is (1 − *c*)*IP^ij^*(*bj*). The sum of these two expressions equals the expression in the logarithm above. The score for all *s* sites in the read is defined as −∑j=1slog⁡(c3+(1−4c/3)PPij(bj)).

Notice that the full phylogenetic likelihood for the entire tree, under standard models of molecular evolution with equal base frequencies and not accounting for errors isℓ=∑j=1slog⁡(∑v∈{A,C,T,G}IPij(v)pvbj(t))

where *p_vbj_*(*t*) is the time dependent transition probability from base *v* to base *b_j_* in time *t*.

This statement takes advantage of the fact that, under time-reversibility, the posterior for a base in an node is proportional to the fractional likelihood of that base in the node, if the tree is rooted in the node. For small values of *t*, ℓ converges to log(*IP^ij^*(*b_j_*)).”

The equation for the phylogenetic likelihood is displayed in equation 7.

P16 Interesting that this classifies N as a gap. Sometimes Ns are ambiguous sequencing calls.

We cannot obtain the IP(*N*), therefore we classify it the same as a gap.

P16 What is m(1)?

Typo fixed.

Figure 2: I found "error/polymorphism" confusing, as at first I thought it was the ratio of the two. Suggest just "error" in the figure and caption, and note in the text that it could be in fact polymorphism. Also, there are lots of triangles for Tronko. Please introduce in the text what the cut-off is. You state that there is a cut-off on P4, and hopefully in one additional sentence you could give an intuition. Font is too small.

We added a sentence to explain ”error/polymorphism”:

We use the term ”error/polymorphism” to represent a simulated change in nucleotide that can be either an error in sequencing or a polymorphism.

We explain in the text what the cut-off means:

”After calculation of scores, the LCA of all of the lowest scoring nodes, using a user-defined cut-off parameter, is calculated. For example, if the cut-off parameter is 0, only the highest scoring node (or nodes with the same score as the highest scoring node) is used to calculate the LCA. If the cut-off parameter is 5, the highest scoring node along with all other nodes within a score of 5 of the highest scoring node are used to calculate the LCA.”

There are many points (triangles) for Tronko because we apply 5 cut-offs (0, 5, 10, 15, and 20). None of the other methods allow a similar comparison therefore all other methods only have a single point.

Figure 3: These "rainbow" schemes are now not in favor compared to color schemes like viridis and variants; see https://journals.plos.org/plosone/article?id=10.1371/journal.pone.0199239

The color scheme that we use is a red to blue color scheme. If we were to use a different color scheme, it would be difficult to see the off-diagonal points.

Figure 6: Please consider color scheme, the yellow is basically invisible.

We changed the color scheme for the majority of the plots.

Figure S4. I'm not sure what "assignment rate" is. Is the fraction of time the program produced any result? Could you label the Tronko lines with Tronko?

Tronko is labeled in this figure. Additionally, we added the following to the text:

”Additionally, in terms of the species assignment rate (the percentage of queries that were assigned at the species level), Tronko assigns the most queries.”

All Figure legends should identify the gene target and data set.Reviewer #3 (Recommendations for the authors):We have divided most of our considerations into three areas, regarding (a) the methods that this ms. introduces, (b) the results that they produce and (c) the Tronko software.(a) The methods take familiar ground as their starting point, and then take a new direction in terms of approximations and shortcuts to implement something resembling a full probabilistic (maximum likelihood) inference technique, but fast enough and with low-enough memory requirement to be practical for modern-sized datasets (reference and sample). We need no convincing that this is a valid approach, and is sufficient for the methods proposed to be of value if they produce good-enough results.That said, there were a few instances of other relevant work not being quoted that I think would be of interest to some readers. We don't think this ms. needs to address all these points in a detailed manner, but where other relevant work has not been cited then readers can reasonably ask whether Pipes and Nielsen are unaware or have simply decided not to incorporate some other existing ideas; or whether they tried to incorporate ideas that ultimately did not work; whether some alternative concepts are somehow incompatible with the approach they did take; or whether there is some other reason. Some specific examples we are curious about and would like to see discussed in this manuscript:* what about methods that combine composition-based (kmer) and phylogenetic approaches, essentially replacing likelihood calculations with kmer statistics in a phylogenetic framework? See work like https://academic.oup.com/bioinformatics/article/35/18/3303/5303992 and others from that group, including recent work by Romashchenko

In our revised manuscript, we add comparisons to two additional phylogenetic placement methods, pplacer and APPLES-2. Both RAPPAS (Linard et al., 2019) and the subsequent EPIK (Romashchenko et al. 2023) are compared to pplacer. RAPPAS, in particular, has worse accuracy than pplacer especially when errors are present (see Figure 4 of Linard et al. 2019). For EPIK, they demonstrate that it is similar in accuracy to pplacer. However, the benchmarking dataset PEWO (Linard at al., 2020) they use is a much smaller dataset (backbone tree of 900 species) than what we use in our manuscript. Additionally, they did not test for different error rates in Romashchenko et al.. (2023). In our revised manuscript, we show that our method is the same or better than pplacer on larger reference databases and with different error rates.

* since the ms. argues so strongly for the benefits of phylogenetic methods, it would be interesting to see how the authors consider Tronko to compare to phylogeny-based methods other than pplacer, for example papara and PAGAN and others e.g. mentioned in https://www.frontiersin.org/articles/10.3389/fbinf.2022.871393/full

Unlike the majority of phylogenetic placement methods, Tronko does not require the query sequences to be aligned with the reference sequences. PaPaRa and PAGAN are both phylogeny-aware alignment programs intended to align the query sequences to the reference sequences. In this manuscript, we show results for different alignment programs (MAFFT and FAMSA) for building the alignment of the reference sequences and Needleman-Wunsch and Wavefront Alignment algorithm for aligning the queries (Figure S6). All choices, withthe exception of Wavefront Alignment on short queries, showed similar results. Additionally, we show results for using different alignment programs, HMMER and MAFFT, for aligning the query sequences to the reference sequences to use with APPLES-2 and pplacer.

(b) The results seem convincing and reasonable metrics were used for evaluation. However,* since the method's highlight is incorporating phylogeny, it would be interesting to see some comparisons of results using a metric that takes phylogeny into account. For example, while calculating accuracy of taxonomic assignment, instead of using a binary score (correct/incorrect), could you calculate a distance between assignment and real value? A measure of how wrong incorrect assignments could be of relevance to many users in terms of biological interpretations. Arguably, a tool that misassigns to close species would be preferable over another that misassigns to further species

We agree that this could be an interesting addition. However, the manuscript is already very large and we are struggling finding place for all figures and results. We consider this, which has so far not been done in the literature, to be a possible subject of future research.

* very poor performances of some tools in mock community analyses might make readers ask if the default parameters of the tools were right or not for this comparison. For example, for Braukmann et al. data the authors say "MEGAN did not assign any reads at the species or genus level"

There was an issue for our pipeline for this analysis. MEGAN had an 15% recall and 0% misclassification rate at the species level and a 50% recall and 0.5% misclassfication rate at the genus level for the Braukmann et al. dataset. We amended Figures 6D and S12 accordingly. For metaphlan2, it is not surprising that the default database does not assign COI reads as it was developed to assign microbial communities.

* some claims about other methods' results being dominated by Tronko based on the convex hull seem a bit overstated, when there has been no investigation of how the other methods' recall/misclassification response curves could appear if other parameter settings were used

We only compared against the default parameter settings of the other methods. (Ye at al. (2019) l/fulltext/S0092-8674(19)30775-5) showed that the biggest differences in results from different species assignment methods come from confounding effects of differences in the reference databases. We also see this, for example, when comparing kraken2-CRUX to kraken2-default for our mock community datasets. In every analysis in this manuscript, we have kept comparisons using a uniform database against all methods.

(c) Tronko software is publically available on GitHub with clear documentation, example files, and working example commands. Nevertheless there is room for improvement for Tronko to be widely utilised by the community:* The software installation and execution can be tested on a few different computers and ideally operating systems. We failed to install the software from the source but Singularity installation works fine.

We have released a.zip package in Github Actions workflow. For Windows and Mac users, we recommend the Singularity installation.

* tronko-build input files are preprocessed with multiple softwares. Although the authors provide example files, there is not a guide on how to create those files for our own gene of interest. A couple of sentences (or maybe more, for users who are not so familiar with the data types and analyses involved) about how those files were prepared would be very helpful.

We provide two example datasets and a guide on how to run the species assignment (https://github.com/lpipes/ tronko?tab=readme-ov-file#tronko-build-example-datasets).

* tronko-build example files include some files with double extensions (e.g..fasta.ann). These are not mentioned in the readme. It is not clear what they are. They are not mentioned in the inputs of the command nor did they appear in the output directory when we ran it.

These files with double extensions are the bwa index files that are used for tronko-assign. The user can choose to build them before assignment or skip the build (with option -6) if the files are already there. The bwa index files are not required as input. We added the following text to the github page https://github.com/lpipes/tronko?tab=readme-ov-file#tronko-assign-usage:

You will also need a FASTA file (not gzipped) of all of your reference sequences in the reference database (use the option -a). tronko-assign will create a bwa index of the reference sequences with the extension of *.fasta.ann, &c. If you already have the bwa index files present in the same directory and naming scheme as your reference sequences, you can choose skip the bwa index build use -6.